# Towards Understanding Continual Factual Knowledge Acquisition of Language Models: From Theory to Algorithm

## Abstract

Continual Pre-Training (CPT) is essential for enabling Language Models (LMs) to integrate new factual knowledge without erasing old. While classical CPT techniques like data replay have become the standard paradigm, the mechanisms underlying how LMs acquire and retain facts over time, termed as continual Factual Knowledge Acquisition (cFKA), remain unclear. In this work, we present a theoretical framework that characterizes the training dynamics of cFKA using a single-layer Transformer with linear attention, offering a unified explanation for the behavior of popular CPT methods. Our analysis reveals that regularization-based methods merely adjust the convergence rate of parameters without altering the inherent forgetting tendency, whereas data replay methods shift convergence dynamics and stabilize pretrained knowledge. Building on these insights, we propose a novel generative data replay approach, called **S**electing **T**okens via attenti**O**n **C**ontribution (STOC), which identifies influential factual snippets to guide replay generation. Extensive experiments on both synthetic and real-world datasets validate our theoretical findings and demonstrate that STOC effectively enhances cFKA by mitigating catastrophic forgetting.

## 1 Introduction

Large language models (LLMs) have acquired abundant factual knowledge during open-domain Pre-Training (PT) (Lin et al.; Wang et al., 2023), yet they still require billions of tokens in Continual Pre-Training (CPT) to adapt to specific downstream domains (called Domain Adaption) or newly emerging knowledge (called Mid-training) (Ke et al., 2023; Yıldız et al., 2024; Han et al., 2020; Wang et al., 2025). Similar to other continual learning settings, while new facts can be learned, the original knowledge tends to be forgotten, referred to as *catastrophic forgetting* (Zheng et al., 2025; Zucchet et al., 2025). Understanding the learning dynamics of continual Factual Knowledge Acquisition (cFKA) (Ou et al., 2025; Wang et al., 2025), i.e., how language models (LMs) memorize new knowledge together with pretrained knowledge during CPT, is crucial to inspire advanced methods.

Several theoretical analyses have been employed to interpret the training dynamics of LMs (Tian et al., 2023; Nichani et al., 2024b; Ren & Sutherland, 2024). However, it's still unclear how traditional continual learning techniques influence the training dynamics of cFKA and mitigate catastrophic forgetting, particularly in the presence of complex textual inputs. In fact, one factual knowledge triplet of (subject, relation, object) can be conveyed through various formats, posing challenges to the unified representation and memorization. See Appendix C for more discussions on related works.

To demystify the dynamics of cFKA, it is important to understand how the learnable parameters evolve during PT and CPT. In this paper, we formally characterize the learning dynamics for a single-layer Transformer with linear attention by assuming Stochastic Gradient Descent (SGD) training where attention modules are tuned with different learning rate (Li et al., 2023; Tian et al., 2023; Chen et al., 2024). Based on these assumptions, we prove several interesting behaviors. In one respect, the factual knowledge is decomposed into frequency-based information and stored at the token level. In another respect, the LM allocates attention scores according to the token diversity of the relevant

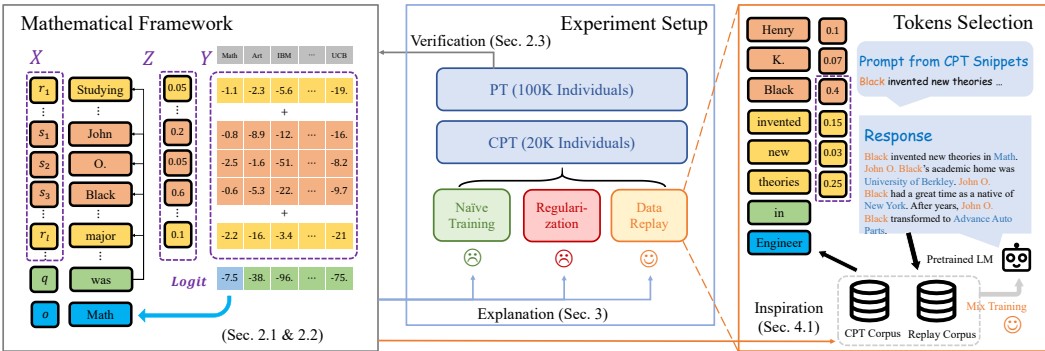

Figure 1: The overall structure of the paper. In Section 2, we present our mathematical framework and main theoretical results, whose effectiveness is verified during controlled PT. After that, we analyze popular continual learning methods in Section 3. Inspired by these findings, we propose a new generative data replay method in Section 4.

answers. Despite simplicity, these findings can explain FKA phenomena observed in multi-layer LMs such as why augmentation helps, indicating their usefulness for interpretation.

Furthermore, the proposed mathematical framework provides a new perspective on understanding the mechanism of popular CPT techniques in cFKA. On one hand, we demonstrate that regularization-based methods can alter merely the convergence rate rather than the convergence point for the cFKA dynamics. Thus, they fail in altering the inherent forgetting tendency in the cFKA scenario. On the other hand, we find that even a small ratio of replay data can eliminate catastrophic forgetting. In addition to modifying association memories, data replay amplifies the oscillation amplitude when convergence is reached, enabling the pretraining knowledge to be retained more significantly.

Based on the analysis, we propose a generative data replay method called **S**electing **T**okens via attenti**O**n **C**ontribution (STOC). STOC selects segments in CPT data through token attention scores. These segments are then used as prompts to generate replay data, so that pretrained LMs tend to generate responses that contain pretrained factual knowledge. Our method mitigates catastrophic forgetting effectively to achieve better performance in both synthetic and real-world experiments.

## 2 THEORETICAL FRAMEWORK FOR FACTUAL KNOWLEDGE ACQUISITION

In this section, we investigate how learnable parameters evolve during training to characterize the LMs' memorizing behaviors. Under assumptions of structured inputs, a single-layer LM with linear attention, and SGD training of different updating rates, we build a theoretical framework and estimate the training dynamics of learnable parameters. Before applying our theoretical framework to cFKA process (in the next section), we first validate its effectiveness on modern multi-layer LMs under a controlled PT environment. This is because PT isolates the LMs from initialization confounders while sharing the same underlying mechanisms as the CPT phase. that our theoretical analysis can provide explanations for several empirical FKA phenomena, e.g., how data augmentation helps LMs generalize across different formats (Allen-Zhu & Li, 2023). For an overview of the notations used in the paper, please check Table 6 in the Appendix B.

### 2.1 INPUT STRUCTURE AND MODEL ARCHITECTURE

**Input Structure** We suppose all the factual knowledge can be represented by "(subject, relation, object)" triples (Paulheim, 2016). Without loss of generality, every subject/relation/object is considered drawn from certain candidate sets. For convenience, we regard each subject can be tokenized into several tokens, each object corresponds to a unique token. For example in Figure 1(left), a person's full name can be divided into first name, middle name, last name token, and his major can be "Math". Besides, we suppose the relation can be conveyed by templates with $L$ tokens, e.g., the relation about major can be "Studying until midnight every day, [Name]'s major was [Major]". Also, we suppose

there is always a unique query token $x_{L+1} = q$ before $x_{L+2} = o$ such as "was". During generation, the query token $q$ acts as the pivot to calculate attention scores for each token.

**Model Architecture** We employ a simplified one-layer transformer as the knowledge learner, such simplification is common in previous works (Tian et al., 2023; Ahn et al., 2023; Nichani et al., 2024a; Zhang et al., 2024b; Nichani et al., 2024b). First, We set $D$ as the vocabulary size and let $x_l \in [D]$ be discrete token, $\boldsymbol{x}_l = \boldsymbol{e}_{x_l} \in \mathbb{R}^D$ be one-hot vector, $\boldsymbol{X} = [\boldsymbol{x}_1, \boldsymbol{x}_2, \dots, \boldsymbol{x}_L]^\top \in \mathbb{R}^{L \times D}$ be the input matrix. Next, word embedding $\boldsymbol{E} \in \mathbb{R}^{D \times d}$ is applied. Then, there is a single-head linear-attention module parameterized by $\boldsymbol{W}_Q, \boldsymbol{W}_K, \boldsymbol{W}_V, \boldsymbol{W}_O \in \mathbb{R}^{d \times d}$, outputing

$$\boldsymbol{a} = \boldsymbol{X} \frac{\boldsymbol{E} \boldsymbol{W}_K \boldsymbol{W}_Q^\top \boldsymbol{E}^\top}{\sqrt{d}} \boldsymbol{x}_{L+1}, \quad \boldsymbol{h} = \boldsymbol{W}_O \boldsymbol{W}_V^\top \boldsymbol{E}^\top \boldsymbol{X}^\top \boldsymbol{a}.$$

Finally, we assume an tied unembedding matrix $\boldsymbol{E}^\top$ is adopted, yielding

$$\mathrm{logit}(o \mid \boldsymbol{X}) = \boldsymbol{x}_o^\top \boldsymbol{E} \boldsymbol{h}, \quad \hat{p}(\cdot | \boldsymbol{X}) = \mathrm{softmax}(\mathrm{logit}(\cdot \mid \boldsymbol{X})).$$

Equivalently, the whole model can be re-parameterized as

$$\boldsymbol{Y} := \boldsymbol{E} \boldsymbol{W}_O \boldsymbol{W}_V^\top \boldsymbol{E}^\top \in \mathbb{R}^{D \times D}, \quad \boldsymbol{Z} := \boldsymbol{E} \boldsymbol{W}_K \boldsymbol{W}_Q^\top \boldsymbol{E}^\top / \sqrt{d} \in \mathbb{R}^{D \times D}.$$

We remark $y_{o,s} = Y_{o,s}$ represents how much token $s$ supports the object $o$ through $\mathrm{logit}(o \mid \boldsymbol{X})$, and $z_s = Z_{s,q}$ determines the non-normalized attention score for token $s$. In the toy example of Figure 1, token "John" contributes to the logit of token "Math" by $y_{o,r_1} = -0.8$ with attention weight $z_{r_1} = 0.2$. Intuitively, $\boldsymbol{Y}$ can be understood as simplified FFN modules storing the acquired knowledge in the modern LMs, which maps subject tokens and relation tokens to object tokens (Geva et al., 2021). $\boldsymbol{Z}$ can be understood as Attention parameters in Transformers (Vaswani et al., 2017), playing a critical role in moving information to the query token for prediction (Sun et al., b).

## 2.2 TRAINING DYNAMICS INDUCED BY SGD

Aligned with practice, Cross-Entropy loss is adopted in the CPT process:

$$\mathcal{L} := -\log \hat{p}(x_{T+2} \mid \boldsymbol{X}) = -\mathrm{logit}(x_{T+2} \mid \boldsymbol{X}) + \log \sum_{x_o} \exp\left(\mathrm{logit}(x_o \mid \boldsymbol{X})\right).$$

As many previous works do (Tian et al., 2023; Li et al., 2023; Chen et al., 2024), we assume the learning rate satisfies $\eta_Y \gg \eta_Z$, so that we consider $z$ to be static when analyzing $\boldsymbol{Y}$'s dynamics. Based on these assumptions, we can derive the approximate evolving dynamics of the parameters, thereby characterizing the model's factual learning behavior. Detailed proof is in the Appendix F.

Assuming attention score remains unchanged, we first analyze the convergence of $\boldsymbol{Y}$. Notice that the loss function $\mathcal{L}$ is convex with respect to $\boldsymbol{Y}$, we therefore reveal a state $\boldsymbol{U} \in \mathbb{R}^{D \times D}$ and then prove that the distance between $\boldsymbol{Y}$ and $\boldsymbol{U}$ consistently decreases before convergence. Denoting multiset $\mathcal{O}_s$ as all object tokens associated with token $s$, $\alpha$ is an initialization scale, let $\overline{\boldsymbol{x}_s} = \mathrm{softmax}(\boldsymbol{u}_s)$ be the "mean" of related objects as

$$\overline{\boldsymbol{x}}_s = \frac{1}{|\mathcal{O}_s|} \sum_{o \in \mathcal{O}_s} \boldsymbol{x}_o, \quad \boldsymbol{u}_s = \log \overline{\boldsymbol{x}}_s + \beta_s \mathbf{1}_D, \quad \beta_s = \alpha - \frac{1}{D} \sum_o \log \overline{\boldsymbol{x}}_{o,s}.$$

**Theorem 1** (Dynamics of $\boldsymbol{Y}$). *Let* $\boldsymbol{\xi}(t) := \boldsymbol{x}_{L+2}(t) - \mathrm{softmax}(\sum_s z_s \delta_s(t) \boldsymbol{u}_s)$ *represents the perturbation term, and let error term after $t$ step updates be* $\boldsymbol{e}_s(t) := \boldsymbol{y}_s(t) - \boldsymbol{u}_s$. *Then using 1-st order Taylor expansion we have the following approximation:*

$$\boldsymbol{e}_s(t) \approx (I - \eta_Y z_s \boldsymbol{H}_s)^t \boldsymbol{e}_s(0) + \eta_Y z_s \sum_{\tau=1}^t (I - \eta_Y z_s \boldsymbol{H}_s)^{t-\tau-1} \boldsymbol{\xi}(t), \tag{1}$$

*where $\delta_s(t)$ is an counter of token $s$, $\boldsymbol{H}_s := \mathrm{diag}(\overline{\boldsymbol{x}}_s) - \overline{\boldsymbol{x}}_s \overline{\boldsymbol{x}}_s^\top$ is the Jacobian matrix.*

Eq. (1) indicates that throughout the gradient flow, $\boldsymbol{Y}$ gradually approaches $\boldsymbol{U}$ and ultimately oscillates around its vicinity, somehow like a damped oscillator with damping. The first term of

Table 1: Performance of the LMs with different augmentation strategies. 5/1/P-Aug represents that one individual corresponds to 5/1/Possion($\lambda = 5$) biographies. Green cell indicates that there is no significant generalization gap, while pink cell indicates a significant generalization gap conversely.

| Aug | Pythia-160m (Train) | | | Pythia-160m (Test) | | | Qwen2.5-0.5B(Train) | | | Qwen2.5-0.5B(Test) | | |
|---|---|---|---|---|---|---|---|---|---|---|---|---|
| | hFTA | sFTA | EM | hFTA | sFTA | EM | hFTA | sFTA | EM | hFTA | sFTA | EM |
| 5-Aug | 95.44 | 93.64 | 81.88 | 94.65 | 92.64 | 81.27 | 95.75 | 94.63 | 83.93 | 95.08 | 93.88 | 83.43 |
| 1-Aug | 95.82 | 95.27 | 8.85 | 15.42 | 14.43 | 2.71 | 96.15 | 95.76 | 51.43 | 13.80 | 12.98 | 2.94 |
| P-Aug | 95.46 | 93.52 | 81.30 | 94.37 | 91.77 | 78.06 | 95.99 | 95.43 | 83.73 | 95.25 | 94.49 | 83.18 |

exponential decay determines the training convergence rate by the largest eigenvalue $\lambda_{\max}^+(\boldsymbol{H}_s)$. For an information-rich token $s$ like "John" in the toy example, $\overline{\boldsymbol{x}}_s$ will be sharp and $\lambda_{\max}^+(\boldsymbol{H}_s)$ will be small. Given the same optimization steps, $\boldsymbol{y}_s$ will exhibit a smaller error and achieve quick convergence, indicating that the LM learns factual knowledge of "John" fast. The second oscillation term remains at a fixed amplitude of $O(\lambda_{\min}^+(\boldsymbol{H}_s))$ as $\boldsymbol{\xi}(t)$ is bounded, constraining the range of $\boldsymbol{Y}$ and therefore determining the error bar at convergence. This term, naturally inherited from SGD training, can serve as a reminder to help LM remember training samples, especially those with low frequency. Also, if "John" rarely appears in the training corpus, then a bigger $\lambda_{\min}^+(\boldsymbol{H}_s)$ will lead to better memory for related knowledge.

In short, $\boldsymbol{Y}$ partitions knowledge into tokens and organizes them by associating each token with its co-occurring object tokens from a frequency perspective. Such token-level knowledge is aggregated through the attention module to provide a guide for object prediction. Theorem 1 can be an umbrella describing the knowledge evolution for each token, leading us to the following conclusion.

> **Fact-to-Frequency Abstraction:** The factual knowledge is decomposed into frequency-based information and stored at the token level. For one certain token, the convergence rate and oscillation amplitude of the knowledge depend on the flatness of its associated tokens.

We then start to analyze how $\boldsymbol{Z}$ changes in the training process. Intuitively, LMs should learn to assign attention scores according to token importance. The following theorem indicates that such importance is measured through the $\ell_2$-norm of $\boldsymbol{y}_s$.

**Theorem 2** (Dynamics of $\boldsymbol{Z}$). *The following quantity remains constant throughout the training:*

$$\frac{d}{dt}\left[\left(\frac{1}{\eta_Z}z_s\right)^2 - \sum_o \left(\frac{1}{\eta_Y}y_{o,s}\right)^2\right] = 0. \tag{2}$$

Theorem 2 reveals the relationship between the values of $z_s$ and $\boldsymbol{y}_s$. Neglecting the influence of the periodic perturbation and set $\boldsymbol{y}_s = \boldsymbol{u}_s$, Eq. (2) specifies a pattern by which the converged model allocates attention scores through a certain Diversity Index (DI, (Simpson, 1949)), defined as

$$z_s = \mathrm{DI}(\overline{\boldsymbol{x}}_s) := \frac{\eta_Z}{\eta_Y}\sqrt{\sum_o (\log \overline{x}_{o,s} + \beta_s)^2} + C, \tag{3}$$

where $C$ is a constant depending on the initialization. For token $s$ whose associations are more balanced such as "the", it serves as a common token with less information, and it gains a lower attention score as the DI is small. In all, we can describe the attention allocation behavior as follows:

> **Diversity-Aware Attention Assignment:** The LM assigns attention scores according to the related-object diversity of each token. Common tokens will receive lower attention scores.

**Remark** Further extensions and discussions on the model architecture are provided in the Appendix D.

### 2.3 DEDUCTIVE EXPERIMENTS: DATA AUGMENTATION HELPS GENERALIZATION

Through several bold assumptions, we have proved several behaviors of the model in FKA. Despite the simplicity of the assumed model architecture, it will be shown in this section to possess strong

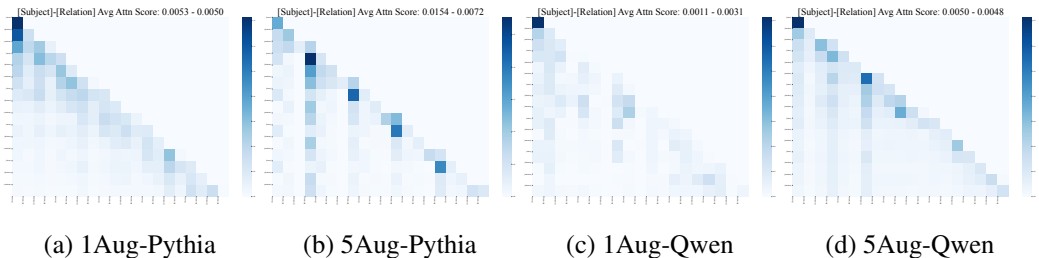

| (a) 1Aug-Pythia | (b) 5Aug-Pythia | (c) 1Aug-Qwen | (d) 5Aug-Qwen |

Figure 2: The attention assignment over different types of tokens in the test biography of the example individual. The assigned attention scores are averaged across tokens within each category (subject or relation). See detailed token-level attention score heatmaps in Appendix H.2, Fig. 4 and 6.

explanatory power even for multi-layer LMs. As a representative example, we illustrate how data augmentation enables LMs to generalize across different knowledge formats. As complementary supports, we provide more interpretation of model behaviors as deductive experiments, e.g. performance plateaus (Zucchet et al., 2025) in Appendix I.1.

**Dataset** We first introduce a synthesis task we used for controlled experiments, which to some extent follows the setup in (Allen-Zhu & Li, 2023). The constructed `Biography` dataset contains 120,000 individuals, each characterized by five relations: birthday, birthplace, university, major, and company. One biography text can be generated by inserting an individual's name and attributes into pre-defined templates. It can be readily verified that such an input format aligns with the assumptions established earlier. To examine the LMs' behavior of FKA during every stage, we then divide the dataset into the PT and CPT corpora at a 5:1 ratio with respect to the number of individuals. See rationale discussion, further details and generated examples in the Appendix D and G.

**Setup** We adopt `Pythia-160M` and `Qwen2.5-0.5B` as our research models for their representative modern LM architectures. In our quest to understand the role of data augmentation, we adopt three different settings for training biography construction: (1) one biography per individual; (2) five biographies per individual; and (3) the number of biographies per individual follows a Poisson distribution of $\lambda = 5$. At the same time, every individual is provided with another three biographies for evaluation, where individual names and templates are used as prompts to generate corresponding attributes. In line with prior works, we employ *hard/soft First Token Accuracy* (hFTA/sFTA) and *Exact Match* (EM) as metrics to investigate training dynamics.

**Behavior** As we can see in Table 1, although all LMs memorize training samples successfully, the LM trained with data augmentation exhibits significantly better performance on the test samples, highlighting the crucial role of data augmentation in improving generalization. It is noteworthy that this phenomenon aligns with the observation of knowledge robustness proposed in (Allen-Zhu & Li, 2023), also observed in real-world experiments across various LLMs and scenarios (Allen-Zhu, 2025).

**Explanation** The preceding theoretical framework suggested a possible mechanism for this phenomenon. Uniform or partial data augmentation leads to a more diverse and abundant set of associated answers for each template. For a certain template token $s$ (also called relation token before), its corresponding $\overline{x}_s$ becomes more uniform once data augmentation is adopted. In this case according to Theorem 2, it will receive a lower attention score and thus encourage the LM to rely more on subject information for prediction. As a result, the model tends to generate outputs that are more consistent across templates, realizing the desired generalization ability. Motivated by the analysis, we investigate the attention matrices of LMs processing the same example data (averaged across layers and attention heads). As depicted in Fig. 2, LMs trained with data augmentation focus more on the individual names according to the average attention scores for subject or relation tokens.

> **Influence of Data Augmentation on Generalization:** Data Augmentation enhances generalization between different formats by altering the diversity index of relation tokens and encouraging LMs to predict according to subject information.

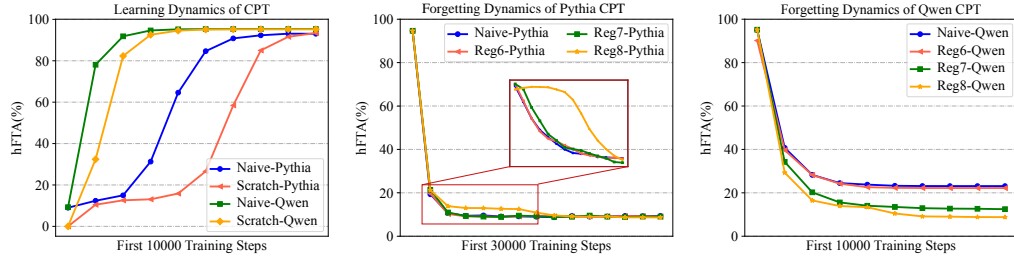

Figure 3: The left figure illustrates the learning dynamics under Naive/Scratch CPT (measured by continual hFTA), while the middle and right show the forgetting dynamics under Naive/Regularization CPT (measured by original hFTA). We perform uniform down-sampling with respect to the step count. All three figures employ Exponential Moving Average with $\alpha = 0.8$) to smooth the noise.

## 3 ANALYSIS ON REGULARIZATION AND DATA-REPLAY BASED CFKA

Under the theoretical framework in the previous section, we now analyze two representative categories of continual learning methods: regularization-based and data replay methods. We discuss their underlying mechanisms by examining how these approaches influence the training dynamics, and thus prove their effectiveness to eliminate catastrophic forgetting. Other types of continual learning methods, such as gradient projection and task vector, are also briefly discussed in the Appendix I.

As a preliminary warm-up, we begin by characterizing LMs' catastrophic forgetting in cFKA in the absence of any continual learning techniques. To begin with, according to the training dynamics of $Y$, the converged LM will be entirely governed by the CPT corpus. While this ensures the model to acquire new factual knowledge successfully, the pretrained knowledge undergoes complete forgetting. Notably, because $Z$ has already escaped from small initialization in the PT stage, the optimization of $Y$ proceeds at a faster rate, accelerating picking up new knowledge while forgetting old. Thus, the pretrained model will learn faster than the LM trained from scratch. As for the training dynamics of $Z$, since LMs have learned to assign attention scores through DI, it changes little once there is little difference between the DI of $\overline{x}_s$ for both subject tokens and relation tokens. Empirical verification is in Figure 3 and Table 2, which indicates the LM learns faster but suffers from catastrophic forgetting.

> **Rapid Acquisition while Rapid Forgetting:** Without any continual learning techniques, the LM will preserve the attention patterns acquired in the pre-training phase. It will learn new knowledge at a faster pace, with the cost of a sudden drop in previously acquired knowledge.

### 3.1 REGULARIZATION FAIL IN SHIFTING PRETRAINED KNOWLEDGE

As one of the most common approaches in continual learning, regularization-based methods work by adding additional constraints to the loss function, so as to prevent LMs' predictions from changing drastically. Although recent studies have shown that the implicit bias of SGD drives the model to converge to the optimal solution closest to the initialization in terms of $\ell_2$ distance (Soudry et al., 2018; Ji & Telgarsky, 2019), regularization-based methods advocate modifying the loss function to weight parameters according to their importance (Zenke et al., 2017; Aljundi et al., 2018), as the loss objective with regularization becomes in the form of

$$\mathcal{L} = \mathcal{L}_{\text{new}} + \frac{k}{2} \sum_i \omega_i (\theta_i - \theta_i^*)^2, \tag{4}$$

where $k$ is the regularization coefficient, $\boldsymbol{\theta}^*$ is the reference parameter before continual training, and $\boldsymbol{w}$ is the importance measure. A typical example is Elastic Weight Consolidation (EWC) (Kirkpatrick et al., 2017), which measures parameter importance through the Fisher Information Matrix. Although regularization has been highly popular for years, its effectiveness has proven to be limited when applied to practical LLM CPT scenarios (Sun et al., 2024; Yang et al., 2023; Shi et al., 2024).

We now provide a theoretical explanation using the framework in the previous section, where the changes induced by the regularization objective are marked in blue. As the additional term in the

Table 2: Performance of the LMs with different regularization coefficients $k$. When $k = 0$, no regularization term is added when calculating objects. Once continual hFTA is up to 90%, we employ an early stopping mechanism to prevent LMs from further forgetting.

| $k$ | Pythia-Original | | | Pythia-Continual | | | Qwen-Original | | | Qwen-Continual | | |
|---|---|---|---|---|---|---|---|---|---|---|---|---|
| | hFTA | sFTA | EM | hFTA | sFTA | EM | hFTA | sFTA | EM | hFTA | sFTA | EM |
| 0 | 9.13 | 8.62 | 1.20 | 94.29 | 92.97 | 68.52 | 25.15 | 22.16 | 5.11 | 95.36 | 94.61 | 65.21 |
| 1e8 | 8.69 | 7.91 | 1.09 | 93.20 | 88.61 | 11.79 | 13.24 | 7.58 | 1.23 | 95.18 | 93.41 | 10.04 |
| 1e7 | 9.10 | 8.61 | 1.18 | 92.68 | 87.91 | 9.66 | 12.63 | 11.40 | 1.79 | 93.40 | 90.10 | 18.59 |
| 1e6 | 9.01 | 8.63 | 1.28 | 93.22 | 89.32 | 39.52 | 23.20 | 24.61 | 4.48 | 95.18 | 93.26 | 39.67 |

form of Eq. 4 is introduced, the updating gradient of $\boldsymbol{Y}$ becomes

$$\dot{y}_{o,s} = \eta_Y z_s \delta_s(t) \left[ \delta(x_{T+2} = o) - \hat{p}(o \mid \boldsymbol{X}) \right] - k\eta_Y w_{o,s}(y_{o,s} - y_{o,s}^{\text{old}}).$$

Then similar to Theorem 1, the dynamics of the error term $\boldsymbol{e}_s(t)$ become

$$\begin{aligned}
\boldsymbol{e}_s(t) \approx & \left( I - k\eta_Y \operatorname{diag}(\boldsymbol{w}_s) - \eta_Y z_s \boldsymbol{H}_s \right)^t \boldsymbol{e}_s(0) \\
& + \eta_Y z_s \sum_{s=1}^{t} \left( I - k\eta_Y \operatorname{diag}(\boldsymbol{w}_s) - \eta_Y z_s \boldsymbol{H}_s \right)^{t-s-1} \boldsymbol{\xi}_s(t) \\
& + k\eta_Y \sum_{s=1}^{t} \left( I - k\eta_Y \operatorname{diag}(\boldsymbol{w}_s) - \eta_Y z_s \boldsymbol{H}_s \right)^{t-s-1} \operatorname{diag}(\boldsymbol{w}_s)(\boldsymbol{u}_s^{\text{old}} - \boldsymbol{u}_s^{\text{new}}).
\end{aligned} \tag{5}$$

Here we ignore the oscillation term at the end of pre-training without compromising the reliability of the conclusion, and $t$ is defined to the beginning of CPT, and so does $\boldsymbol{H}_s$. The first two terms in Eq. 5 determine the convergence rate by changing the eigenvalues $\lambda_{\max}^+(k\operatorname{diag}(\boldsymbol{w_s}) + z_s\boldsymbol{H_s})$ and $\lambda_{\min}^+(k\operatorname{diag}(\boldsymbol{w_s}) + z_s\boldsymbol{H_s})$. As the amplitude of oscillation increases, the model's convergence speed becomes limited. Moreover, the third term determines the LMs' retention of both PT and CPT knowledge after convergence, and adjusting the value of $k$ enables a trade-off between the two. However, the value of this term is constrained by the smallest positive eigenvalue $\lambda_{\min}^+(\operatorname{diag}(\boldsymbol{w}_s)) = \min_o w_{o,s}$, which indicates that the old knowledge regarding token $s$ can be retained only if each component of $\boldsymbol{y}_s$ is of substantial importance. Considering the relationship between parameter count and knowledge amount (Allen-Zhu & Li, 2024), such a situation will not occur in the FKA scenario. Therefore in the cFKA setting, regularization methods do not alter the convergence point of the model but only affect its convergence rate.

Continuing with the experimental setup introduced in Section 2.3, we apply the regularization method during CPT. First, as demonstrated in Figure 3, EWC does alter the forgetting rate, with a larger $\alpha$ resulting in a more significant deceleration. Second, in the Table 2, although forgetting is slowed down in the early stage of CPT, the final state of the old knowledge remains unchanged, and an excessively large regularization term may even further impair the model's knowledge.

> **Slower but Still Forgetting:** Introducing a regularization term enables the model to forget original knowledge slowly, but it almost fails in mitigating forgetting.

### 3.2 DATA REPLAY STABILIZE PRETRAINED KNOWLEDGE

Besides adding regularization terms, another commonly used approach is data replay, which addresses catastrophic forgetting by storing or generating part of PT data and mixing it together with CPT data. Some literature has discussed the significance of data replay techniques and has further summarized scaling laws to guide data mixing (Wang et al., 2025). An interesting observation is that even when replay data accounts for only a small fraction of CPT samples, it can still effectively alleviate the forgetting of prior knowledge (Gu et al., 2024).

The approximation proposed in the previous section provides a solid explanation for the effectiveness of data replay. Let $\alpha$ denote the percentage of CPT data, the frequency based prediction changes

$$\overline{\boldsymbol{x}}_s = \frac{1 - \alpha}{|\mathcal{O}_s^{\text{old}}|} \sum_{o \in \mathcal{O}_s^{\text{old}}} \boldsymbol{x}_o + \frac{\alpha}{|\mathcal{O}_s^{\text{new}}|} \sum_{o \in \mathcal{O}_s^{\text{new}}} \boldsymbol{x}_o.$$

Table 3: Performance of the `Pythia-160M` with different data replay strategies. $0.5 - 0.9$ denotes the mixing ratio of continual corpus. $+$ represents that the first 6 layers are frozen during training. When the sFTA in continual learning exceeds 90%, the best results in mitigating catastrophic forgetting are highlighted in **bold**, while the second-best are underlined. As ALL and HALF use real pretraining data as replay data, their results are regarded as upper bounds, denoted by grey.

| Target | Replay | 0.5 | | | 0.67 | | | 0.8 | | | 0.9 | | |
|---|---|---|---|---|---|---|---|---|---|---|---|---|---|
| | | hFTA | sFTA | EM | hFTA | sFTA | EM | hFTA | sFTA | EM | hFTA | sFTA | EM |
| Continual | ALL | 94.22 | 91.08 | 64.29 | 94.39 | 91.91 | 66.4 | 94.38 | 91.89 | 70.09 | 94.51 | 91.22 | 70.14 |
| | HALF | 94.39 | 91.79 | 63.44 | 94.49 | 92.27 | 68.35 | 94.52 | 93.1 | 73.01 | 94.51 | 92.88 | 68.8 |
| | LAMOL | 94.19 | 92.58 | 85.33 | 94.35 | 92.94 | 86.84 | 94.38 | 93.13 | 87.29 | 94.08 | 92.67 | 79.76 |
| | LAMOL$^+$ | 94.34 | 92.47 | 82.5 | 94.43 | 92.62 | 81.01 | 94.51 | 93.13 | 78.56 | 94.33 | 92.06 | 75.03 |
| | STOC | 93.65 | 90.47 | 71.41 | 94.11 | 91.48 | 76.62 | 94.15 | 91.46 | 77.04 | 94.17 | 92.13 | 77.43 |
| | STOC$^+$ | 93.92 | 90.29 | 70.65 | 94.4 | 92.04 | 74.24 | 94.33 | 92.11 | 75.73 | 94.43 | 91.96 | 74.93 |
| Original | ALL | 91.48 | 86.56 | 63.33 | 90.13 | 83.81 | 53.18 | 85.6 | 77.85 | 48.32 | 48.55 | 36.52 | 17.44 |
| | HALF | 67.34 | 64.01 | 34.96 | 66.13 | 62.5 | 33.54 | 63.52 | 59.06 | 31.9 | 47.16 | 41.09 | 17.27 |
| | LAMOL | 20.98 | 19.9 | 5.95 | 20.92 | 19.8 | 6.01 | 20.45 | 19.29 | 5.75 | 12.89 | 12.15 | 2.97 |
| | LAMOL$^+$ | 25.39 | 22.18 | 9.29 | 23.14 | 21.69 | 9.47 | 22.78 | 21.33 | 9.22 | 20.06 | 18.88 | 7.58 |
| | STOC | 55.91 | 51.54 | 29.84 | 54.86 | 50.39 | 29.64 | 49.78 | 45.4 | 25.17 | 22.03 | 19.7 | 7.18 |
| | STOC$^+$ | **59.07** | **54.33** | **33.89** | **58.7** | **53.8** | **32.83** | **55.68** | **50.67** | **29.84** | **44.82** | **40.54** | **21.62** |

In particular, the first term in the formula ensures that the pretrained knowledge is retained in the model parameters, with the strength of this knowledge being modulated by the parameter $1 - \alpha$. At the same time, we remark that the amplitude of the oscillation term will become larger, controlled by $\lambda_{\min}^+(\boldsymbol{H}_s), \boldsymbol{H}_s = \mathrm{diag}(\overline{\boldsymbol{x}}_s) - \overline{\boldsymbol{x}}_s \overline{\boldsymbol{x}}_s^\top$. Under the combined effects of these two components, data replay can substantially mitigate the forgetting of pretrained knowledge even when $\alpha$ is large.

Following the experimental setup introduced in the previous section, we construct CPT corpora with varying proportions of replay data to examine how the model learns new knowledge while forgetting old. We consider the following two replay data selection rules: (1) for each individual, one biography is retained as replay data; (2) half of the individuals retain two biographies as replay data, while the other half have no replay data. From the results in Table 3, we observe that data replay plays a crucial role in alleviating catastrophic forgetting. Even when the proportion of replay data is as small as 10%, the model is still able to retain a substantial amount of pre-training knowledge. On the other hand, although the two replay strategies yield a comparable number of replay tokens and the overall replay ratio remains identical, the first replay strategy leads to noticeably less forgetting. These findings imply that replay datasets should encompass a wide range of factual knowledge.

For various reasons like intellectual property protection, storing past data is not always feasible. Generative data replay, therefore, aims to train an auxiliary generative model to produce pseudo-samples as a substitute for the original past data. Specifically, when the target model itself is a generative model, many studies attempt to directly decode replay data from the target model (Sun et al., a). For Example, LAMOL (Sun et al., a) uses special tokens as prompts to have the language model generate replay data. The mechanism of these methods is similar to that of stored data replay, but it places greater demands on the pretrained model to generated original knowledge. Empirical validation can be found in Tabel 3 and Appendix H.2 Table 8.

> **Data Replay Alters Convergence and Amplifies Confidence:** Data replay shifts the convergence point of parameters to retain pretrained knowledge. It also amplifies the oscillations to strengthen the pretrained knowledge confidence.

## 4 DYNAMICS-INSPIRED GENERATIVE DATA REPLAY METHOD

From the above analysis, we validate the crucial effectiveness of data replay in alleviating catastrophic forgetting. However, considering that storage-based replay is largely infeasible in the context of LLMs, where proprietary PT corpora are seldom publicly available, generative data replay still holds vast application potential. Nevertheless, under synthetic experimental settings, we still include storage-based data replay as an upper bound for performance comparison.

Table 4: Comparison of STOC with existing methods on three real-world datasets, measured by averaged soft token accuracy. The error bars are calculated from five sets of random seed training. The LMs are trained with different freezing layers and the best results are reported. If the model's accuracy on new knowledge falls below 95% of the Naive baseline, we consider it fails to acquire enough new knowledge as expected, marked in gray.

| | Method | ZSRE | | Wiki_Bio | | Wiki_Recent | |
|---|---|---|---|---|---|---|---|
| | | Original | Continual | Original | Continual | Original | Continual |
| Pythia | Naive | $24.42_{\pm 0.27}$ | $48.48_{\pm 0.21}$ | $13.22_{\pm 0.12}$ | $32.21_{\pm 0.20}$ | $18.10_{\pm 0.21}$ | $20.39_{\pm 0.25}$ |
| | LAMOL ($\alpha = 0.5$) | $24.48_{\pm 0.22}$ | $47.56_{\pm 0.29}$ | $22.31_{\pm 0.15}$ | $31.33_{\pm 0.17}$ | $16.32_{\pm 0.21}$ | $19.27_{\pm 0.20}$ |
| | LAMOL ($\alpha = 0.8$) | $24.95_{\pm 0.32}$ | $47.12_{\pm 0.21}$ | $20.46_{\pm 0.20}$ | $31.54_{\pm 0.26}$ | $16.16_{\pm 0.24}$ | $17.35_{\pm 0.16}$ |
| | STOC ($\alpha = 0.5$) | $26.88_{\pm 0.23}$ | $47.94_{\pm 0.31}$ | $22.89_{\pm 0.24}$ | $28.05_{\pm 0.19}$ | $17.58_{\pm 0.13}$ | $19.23_{\pm 0.17}$ |
| | STOC ($\alpha = 0.8$) | $\mathbf{27.56}_{\pm \mathbf{0.20}}$ | $47.23_{\pm 0.19}$ | $\mathbf{23.86}_{\pm \mathbf{0.10}}$ | $31.88_{\pm 0.16}$ | $\mathbf{19.36}_{\pm \mathbf{0.13}}$ | $19.56_{\pm 0.19}$ |
| Qwen2.5 | Naive | $34.58_{\pm 0.16}$ | $63.28_{\pm 0.28}$ | $32.33_{\pm 0.16}$ | $35.50_{\pm 0.13}$ | $19.28_{\pm 0.14}$ | $28.42_{\pm 0.18}$ |
| | LAMOL ($\alpha = 0.5$) | $37.54_{\pm 0.19}$ | $58.37_{\pm 0.22}$ | $31.29_{\pm 0.22}$ | $34.49_{\pm 0.23}$ | $20.48_{\pm 0.17}$ | $27.19_{\pm 0.21}$ |
| | LAMOL ($\alpha = 0.8$) | $36.71_{\pm 0.23}$ | $57.44_{\pm 0.17}$ | $34.67_{\pm 0.26}$ | $34.64_{\pm 0.17}$ | $20.15_{\pm 0.22}$ | $27.85_{\pm 0.19}$ |
| | STOC ($\alpha = 0.5$) | $37.12_{\pm 0.17}$ | $62.26_{\pm 0.12}$ | $\mathbf{35.57}_{\pm \mathbf{0.06}}$ | $35.46_{\pm 0.10}$ | $\mathbf{21.40}_{\pm \mathbf{0.11}}$ | $28.75_{\pm 0.16}$ |
| | STOC ($\alpha = 0.8$) | $\mathbf{37.47}_{\pm \mathbf{0.14}}$ | $62.59_{\pm 0.18}$ | $35.28_{\pm 0.13}$ | $33.16_{\pm 0.05}$ | $20.12_{\pm 0.15}$ | $27.34_{\pm 0.18}$ |

In this section, we further propose a dynamics-inspired generative data replay method on the cFKA scenario, which can be naturally derived from our above analyses. We then conduct experiments to evaluate the effectiveness of the proposed method in both synthesis and real-world datasets.

## 4.1 STOC: SELECTING TOKENS VIA ATTENTION CONTRIBUTION

Although data replay is highly effective in mitigating catastrophic forgetting, these methods are still far from reaching their full potential. In particular, existing data replay approaches do not exploit the architectural characteristics of autoregressive Transformer models, leaving room for improvement in terms of the knowledge quality and diversity of the generated samples (Zheng et al., 2025; Huang et al., 2024). Our method aims to automatically build more prompts that enable LMs to generate responses containing pretrained factual knowledge.

As discussed in Sec. 2.3, LMs tend to perform diversity-aware attention assignment. If a token can greatly narrow down the range of correct answers, i.e. the factual knowledge associated with the token is more specific, then it receives a higher attention score. The converse also holds. In this case, the attention score assigned to a token can be used to estimate how much factual knowledge it carries, allowing us to select certain text snippets to prompt the pre-trained LMs to generate replay data.

Based on these insights, we propose a generative data replay method termed as **S**electing **T**okens via attenti**O**n **C**ontribution (STOC). First, for a given piece of CPT example, STOC performs a forward pass to obtain the attention scores of each token, which are then aggregated by averaging across different layers and attention heads. Then, these attention scores are used to select fixed-length snippets from the training example, which can be implemented using a sliding window. Subsequently, the selected tokens serve as prompts to guide the pretrained LMs in producing replay data. Finally, a data selection process can be optionally performed to filter out low-quality data. Further details and some generated examples through STOC are provided in Appendix H.1.

## 4.2 EXPERIMENTS AND ANALYSIS

To evaluate the effectiveness of STOC, we first implement it on the `Biography` dataset, where the experiment setup follows the experiments before. Further details can be found in Appendix H.1. For a fair comparison, we maintain the same token count in the filtered replay data across all methods, instead of applying a data quality score threshold. When the LMs' sFTA on CPT knowledge exceeds 90%, we report the averaged results in Table 3 and 8.

As we can see, STOC successfully mitigates catastrophic forgetting in cFKA scenario and outperforms LAMOL, indicating that STOC can generate higher-quality replay data. Also, STOC can be integrated with other independent continual learning techniques, such as Freezing (Zheng et al., 2025), to further enhance performance, demonstrating its wide applicability. We list several replay biographies generated by STOC as case studies, which can be found in Appendix H.1.

Table 5: Comparison of STOC with existing methods on most popular datasets. The LMs are trained with different freezing layers and the best results are reported. The model responses are generated through prompts with 5-shot examples. The best results are highlighted in **bold**.

| | Method | MMLU | | MMLU-Redux-2.0 | | SuperGPQA | |
|---|---|---|---|---|---|---|---|
| | | Original | Continual | Original | Continual | Original | Continual |
| 0.6B | Naive | 22.49 | 24.40 | 21.93 | 24.39 | 9.48 | 10.98 |
| | LAMOL | 38.87 | 29.42 | 39.04 | 28.05 | 10.60 | 13.87 |
| | STOC | **40.17** | **30.92** | **40.26** | **32.93** | **10.76** | **15.24** |
| 1.7B | Naive | TBD | TBD | TBD | TBD | TBD | TBD |
| | LAMOL | TBD | TBD | TBD | TBD | TBD | TBD |
| | STOC | TBD | TBD | TBD | TBD | TBD | TBD |

To further illustrate the practical utility of STOC, we use `KnowEdit`, a model-editing benchmark, to fine-tune the pretrained base model of Pythia-160M and Qwen2.5-0.5B. Specifically, we select ZSRE, Wiki_Bio, and Wiki_Recent within KnowEdit to conduct our experiments. Detailed description to the setup can be found in Appendix H.1. To maintain consistency with the previous sections, we employ soft average token accuracy on both original and continual knowledge. Such metrics are also conceptualized as *Effectiveness* and *Locality* in the context of knowledge editing tasks.

The results are positioned in Table 4. It can be confirmed that STOC is not only capable of eliminating forgetting, but also ensuring cFKA well when adopted on practical pretrained LMs. According to the generated examples in Appendix H.1, STOC is able to induce high-quality replay data that contains relevant knowledge, reminding the model of old knowledge during CPT.

To further evaluate the effectiveness and scalability of our approach, we conduct experiments on larger-scale and more heterogeneous real-world corpora. We first select `law_judiciary` subset of `IndustryCorpus2` as our CPT source, allowing us to scale domain-specific data to several billion tokens. We additionally use `MMLU`, `MMLU-Redux-2.0`, and `SuperGPQA` as the evaluation benchmark, since answering these questions requires complicated, highly non-structured knowledge rather than simple factual triples. Concretely, we sample 1B training tokens, and use the law subsets (denoted as "continual" to maintain consistency) to assess how much new legal knowledge the model acquires during CPT. The other subsets (denoted as "original" likely) serve to measure knowledge retention, as the pretrained model has already learnt general knowledge. For convinience to obtain pre-trained models of suitable scale, we selecte the Qwen3 Family in the experiment. More details can be found in Appendix H.1.

As shown in Table 5, STOC not only outperforms the baseline methods in mitigating catastrophic forgetting consistently, but also shows clear gains on the continual subsets. A plausible explanation is that replay stabilizes the model's internal representations by maintaining exposure to previously learned distributions. When scaling the continual pretraining data to larger corpora, STOC's improvements remain stable rather than diminishing, indicating its robustness under larger-scale training.

> **STOC:** Selecting Tokens via attentiOn Contribution helps eliminate catastrophic forgetting in both synthesis and real-world scenario, reinforcing the deductibility of our theoretical analysis.

## 5 CONCLUSION

This paper aims to explain LMs' behavior in continual FKA, where new knowledge is learned while pretrained knowledge is prone to being forgotten. To facilitate the analysis, we first design a simplified single-layer Transformer and investigate the training dynamics of its learnable parameters, thereby explaining and empirically validating several phenomena in FKA. Then, based on the above analysis, we shed light on the mechanism of popular continual learning methods such as regularization and data replay, drawing conclusions that align with many existing observations. Finally, inspired by the analyses and observations, we propose STOC, a simple generative data replay method. Extensive experiments demonstrate its effectiveness in eliminating catastrophic forgetting. Further discussion on extensions, limitations, and future work is provided in Appendix D.

## 6 REPRODUCIBILITY STATEMENT

We have made every effort to ensure that our results are fully reproducible. In Section 2.3 and Appendix H.1, we elaborate on the key details of constructing the synthetic dataset. Real-world datasets, pretrained LM checkpoints, and configurations used in our experiments are publicly available, as stated in Appendix H.1. For theoretical proof, we list the main results and corresponding proof in Appendix F. We also provide detailed descriptions of our training procedures, hyperparameters, and evaluation protocols in Section 2.3, Appendix H.1. We ensure all code to reproduce the main results is publicly available at the anonymous website: `https://anonymous.4open.science/r/continual_Factual_Knowledge_Acquision-63B1`.

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

## A    STATEMENTS OF LLM USAGE

In the preparation of this manuscript, Large Language Models (LLMs) were used in a limited but supportive capacity. First, LLMs assisted in improving the clarity, fluency, and readability of the written text by suggesting alternative phrasings and polishing grammar and style, while we guarantee that the substantive content and intellectual contributions remain the responsibility of the authors. Second, in the synthetic data experiments, the LLM was employed to generate candidate templates or perform data augmentation by rewriting, which have been described in the main paper. We ensure all the generated content is carefully reviewed and selected by humans to ensure that they contained no potentially harmful information. Importantly, LLMs were not used for research ideation, data analysis, or producing research findings. All conceptual development, methodological design, and substantive contributions are solely attributable to the authors.

## B    NOTATION TABLE

Table.6 gives the notations of the main quantities in the paper.

Table 6: Overall notation table of the main symbols in the paper.

| | | |
|---|---|---|
| **Basic Notations** | | |
| $t$ | $\mathbb{Z}_+$ | Training Step, the initial moment for PT/CPT is 0 |
| $D$ | $\mathbb{Z}_+$ | Vocabulary Size |
| $d$ | $\mathbb{Z}_+$ | Hidden size of the language model |
| $L$ | $\mathbb{Z}_+$ | Input sequence (prompt) length |
| $s$ | $[D]$ | One token in the vocabulary |
| $x_l$ | $\mathbb{R}$ | The $l$-th token in the input sequence |
| $\boldsymbol{x}_l$ | $\mathbb{R}^D$ | One hot vector corresponding to token $x_l$ |
| $x_{L+1} = q$ | $\mathbb{R}$ | The unique query token to calculate attention scores. |
| $x_{L+2}$ | $\mathbb{R}$ | The object token, ground truth for training |
| $\boldsymbol{X}$ | $\mathbb{R}^{L \times D}$ | One input sequence |
| $\delta_s(t)$ | $\mathbb{Z}_+$ | The number of token $s$ in the training sequence at time $t$ |
| $\mathcal{O}_s$ | / | The multiset of objects associated with token $s$ during training |
| **Learnable Parameters** | | |
| $\boldsymbol{E}$ | $\mathbb{R}^{D \times d}$ | The embedding matrix and tied unembedding matrix |
| $\boldsymbol{W}_Q, \boldsymbol{W}_K$ | $\mathbb{R}^{d \times d}$ | The parameters for attention score calculation |
| $\boldsymbol{W}_V, \boldsymbol{W}_O$ | $\mathbb{R}^{d \times d}$ | The parameters for hidden state calculation |
| $\boldsymbol{Y}$ | $\mathbb{R}^{L \times D}$ | Equivalent reparameterization results, $\boldsymbol{Y} = \boldsymbol{E}\boldsymbol{W}_O\boldsymbol{W}_V^\top\boldsymbol{E}^\top$ |
| $\boldsymbol{Z}$ | $\mathbb{R}^{L \times D}$ | Equivalent reparameterization results, $\boldsymbol{Z} = \boldsymbol{E}\boldsymbol{W}_K\boldsymbol{W}_Q^\top\boldsymbol{E}^\top/\sqrt{d}$ |
| $y_{o,s}$ | $\mathbb{R}$ | The element of $\boldsymbol{Y}$ in row $o$ and column $s$ |
| $\boldsymbol{y}_s(t)$ | $\mathbb{R}^D$ | The $s$-column of $\boldsymbol{Y}$ at time $t$ |
| $z_s$ | $\mathbb{R}$ | The element of $\boldsymbol{Z}$ in row $s$ and column $q$ |
| **Hyperparameters** | | |
| $\eta_Y, \eta_Z$ | $\mathbb{R}$ | Learning rate for parameter $\boldsymbol{Y}, \boldsymbol{Z}$ respectively |
| $\epsilon$ | $\mathbb{R}$ | The initialization scale |
| $k$ | $\mathbb{R}$ | The regularization coefficient |
| $\boldsymbol{w}_s$ | $\mathbb{R}^D$ | The importance weight for $\boldsymbol{y}_s$ |
| $\alpha$ | $\mathbb{R}$ | The percentage of CPT data |
| **Intermediate Variables** | | |
| $\overline{\boldsymbol{x}}_s$ | $\mathbb{R}^D$ | Average of the related object tokens corresponding to token $s$ |
| $\boldsymbol{u}_s$ | $\mathbb{R}^D$ | The convergence center of $\boldsymbol{y}_s$ |
| $\beta_s$ | $\mathbb{R}^D$ | The Normalization coefficient for $\boldsymbol{u}_s$ |
| $\boldsymbol{\xi}_s(t)$ | $\mathbb{R}^D$ | Perturbation term from label-prediction difference at time $t$ |
| $\boldsymbol{e}_s(t)$ | $\mathbb{R}^D$ | The error term between $\boldsymbol{y}_s(t)$ and $\boldsymbol{u}_s$ |
| $\boldsymbol{H}_s$ | $\mathbb{R}^{D \times D}$ | Jacobian matrix for token $s$ |
| $\lambda_{\max}^+(\cdot), \lambda_{\min}^+(\cdot)$ | $\mathbb{R}^{D \times D} \mapsto \mathbb{R}_+$ | The largest/smallest eigenvalue |

## C    RELATED WORKS

Prior research can generally be divided into three aspects: *Catastrophic Forgetting Mechanism*, *Continual Pretraining Methods*, and *Factual Knowledge Acquisition Phenomena*. We primarily focus on the parts related to autoregressive LMs implemented via neural networks.

**Catastrophic Forgetting Mechanism**    Since catastrophic forgetting in neural networks was first reported in McCloskey & Cohen (1989), many studies have endeavored to explain it from a mechanistic perspective. Start from a linear regression model (Evron et al., 2022), research on forgetting states has gradually extended to more complex models (Lee et al., 2021) and complex task ordering (Ding et al., 2024). Recently, another line of methods attempts to measure the internal forgetting state using probing techniques (Davari et al., 2022; Chen et al., 2023). These studies have identified differences in catastrophic forgetting across different layers of neural networks (Wu et al., 2022). These studies laid the foundation for analyzing continual learning mechanisms and have inspired subsequent research on continual learning in Transformer architectures.

**Continual Pretraining Methods**    Most of the current methods for alleviating forgetting during the continuous learning of large language models center on data replay or parameter constraints. LAMOL (Sun et al., a) is a generative data replay method that prompts the model to generate samples of previous tasks by inserting special tokens during training. Another generative approach, HMI-LAMOL  (Maekawa et al., 2023), builds upon it by using hippocampal memory indexing to enhance the generative replay. EWC  (Sun et al., a) measures the importance of model parameters using the Fisher Information Matrix and adopts a regularization loss to constrain updates to critical parameters for previous tasks. MAS (Aljundi et al., 2018) computes the parameter importance based on the sensitivity of the network's output function to parameter changes. GPM (Saha et al.) obtains the Core Gradient Space (CGS) via SVD and constrains updates to the orthogonal subspace to avoid interfering with past knowledge. Zheng et al. (2025) freeze the bottom layers of the model (e.g., input embedding layers) to prevent task alignment from being disrupted. Although these methods incorporate distinctive designs, they do not explicitly leverage the unique characteristics of Transformer architectures, leaving substantial room for further improvement.

**Factual Knowledge Acquisition Phenomena**    Pretrained language models are capable of complex tasks, leading to the belief that they have learned a wealth of factual knowledge (Zhao et al., 2023). However, for a specific piece of fact, whether it has been stored is unclear due to the intricacy of pretraining corpora. To answer this question, Allen-Zhu & Li (2023) firstly propose forming empirical conclusions with fully synthetic data and construct `Biography` dataset. Since then this experimental setup has been widely adopted, leading to a series of conclusions about FKA: (1)After applying data augmentation strategies, the model is able to learn all factual knowledge during pretraining, which remains intact during instruction tuning (Allen-Zhu & Li, 2023). (2)The amount of factual knowledge a model can store is proportional to its parameter count (Allen-Zhu & Li, 2024). (3)If the parameter is insufficient, the model prioritizes storing higher-frequency knowledge (Gu et al., 2025). (4)The knowledge stored is fragile when CPT without data replay (Zheng et al., 2025; Zucchet et al., 2025). These works provide reliable empirical findings, yet still lack a solid theoretical explanation.

## D    DISCUSSION AND FUTURE WORK

**Rationale for Using a Synthetic Dataset.**    The biggest challenge in studying cFKA lies in ensuring the independence of the knowledge contained in the training datasets. Firstly, the CPT dataset should not include content related to PT knowledge, otherwise the evaluation of old knowledge forgetting would be biased. Secondly, if a data replay strategy is introduced, it is necessary to ensure that the replay dataset indeed contains pre-training knowledge. Since real-world datasets rarely come with annotated knowledge content, it is challenging to satisfy the two requirements above. Thirdly, the training data used in real experiments is often extensive, making it infeasible to conduct such controllable experiments at that scale. By constructing the synthetic `Biography` dataset, we fulfill the three requirements above, which provides a solid guarantee for the feasibility of our experiments.

**Potential Extensions of the Proposed Model**    We acknowledge that the mathematical model proposed in Sec. 2.1 differs significantly from the modules in modern LMs. However, we note that

this mathematical LM can be extended in several aspects without affecting the main conclusions:

• *Loss Computation on Every Token* From the analysis in Sec. 2.2, it can be observed that, the output length $L$ does not significantly influence convergence. Therefore, computing the loss for each token is equivalent to executing our data input $L$ times, each with a distinct sequence length.

• *Positional Embedding* Though our main results do not take positional encoding into account, incorporating a fixed relative positional bias term (T5 (Raffel et al., 2020), ALiBi Press et al. (2021)), is straightforward. Specifically, Positional Embedding influences the convergence rate of $Y$ by altering $z_s$ into $z_s + p_s$, but it does not change the convergence point. It can also be verified that a conserved quantity similar to Eq. 2 still exists, i.e., an additional bias term is added to the original DI.

• *Multi-Head Attention* By partitioning all subject tokens and relation tokens and assigning them to different attention heads according to the partition, it is also possible to achieve complete knowledge memorization (Nichani et al., 2024b). For our model, partitioning attention heads causes $Y$ to become a block diagonal matrix. In this case, permuting the tokens is equivalent to performing a congruence transformation on $Y$. Under the assumption of knowledge sparsity, which is a necessary condition given the sufficient parameters assumption, the convergence behavior of $Y$ remains unchanged.

# E    LIMITATION

**Data and Experiments**    Firstly, many of our empirical experiments were conducted on synthetic data, and some conclusions were also validated in this setting. Although the synthetic data pipeline was carefully designed, there still exists a gap between synthetic and real-world datasets. In the Discussion D, we elaborate on the rationale for using synthetic data and argue for the generalizability of our conclusions. Secondly, due to the cost associated with model training, we were constrained to conduct experiments only on LMs with less than 10B parameters, especially when models for large-scale controllable experiments are restricted to 1B parameters or fewer. While the chosen model architectures are thoughtfully selected for representativeness, they still cover only a limited portion of the design space. These decisions, while pragmatic, could constrain the scope over which our conclusions can be reliably applied.

**Hypothesis**    Many of the conclusions derived from our mathematical model were not directly validated through small-scale toy experiments. Instead, we chose to verify them on real multi-layer LMs. Our purpose was to assess the feasibility of the theoretical analysis on practical LMs, at the cost of losing a direct approach to validate the soundness of our assumptions. As compensation, we design several deductive experiments and propose a novel generative data replay method based on our analytical results. These outcomes, in turn, provide evidence supporting our theoretical analysis.

**Theoretical Analyses**    The proposed theoretical analysis may not fully capture the behavior of multi-layer LMs. The biggest difficulty lies in explaining how multiple tokens are combined to form high-level concepts, which is beyond the scope of this paper. For instance, freezing lower-layer parameters has proven to be an effective strategy for mitigating catastrophic forgetting (Zheng et al., 2025). In addition, *Attention Sink* (Xiao et al.) is also lacking an explanation in this paper.

Despite these limitations, we believe this paper provides valuable insights into the subject matter. We will strive to address these limitations through more exploration and refinement in future work.

# F    NOTES AND PROOF ON THEORETICAL ANALYSIS

## F.1    PROOF OF THEOREM 1

**Theorem\* 1** (Dynamics of $Y$). *Let $\boldsymbol{\xi}_s(t) := \boldsymbol{x}_{L+2}(t) - \mathrm{softmax}(\sum_s z_s \delta_s(t) \boldsymbol{u}_s)$ represents the perturbation term, and let error term after $t$ step updates be $\boldsymbol{e}_s(t) := \boldsymbol{y}_s(t) - \boldsymbol{u}_s(t)$. Then using 1-st order Taylor expansion we have the following approximation:*

$$\boldsymbol{e}_s(t) \approx (I - \eta_Y z_s \boldsymbol{H}_s)^t \boldsymbol{e}_s(0) + \eta_Y z_s \sum_{s=1}^{t} (I - \eta_Y z_s \boldsymbol{H}_s)^{t-s-1} \boldsymbol{\xi}_s(t), \tag{6}$$

*where $\delta_s(t)$ is an counter of token $s$, $\boldsymbol{H}_s := \mathrm{diag}(\overline{\boldsymbol{x}}_s) - \overline{\boldsymbol{x}}_s \overline{\boldsymbol{x}}_s^\top$ is the Jacobian matrix.*

*Proof.* First, we remark that the gradient of $Y$ is

$$y_{o,s}(t+1) - y_{o,s}(t) = \eta_Y z_s \delta_s(t) \left[ \delta(x_{T+2} = o) - \hat{p}_t(o|X) \right].$$

Rewrite the above into a recurrence relation for $e_s$ and Substitute $\boldsymbol{\xi}_s(t)$ we have

$$\boldsymbol{e}_s(t+1) - \boldsymbol{e}_s(t) \approx \eta_Y z_s \delta_s(t) \left[ \boldsymbol{\xi}_s(t) + \text{softmax}\left( \sum_s z_s \delta_s(t) \boldsymbol{u}_s \right) - \text{softmax}\left( \sum_s z_s \delta_s(t)(\boldsymbol{e}_s(t) + \boldsymbol{u}_s) \right) \right].$$

For all training data gradients, perform a first-order Taylor expansion of the softmax function,

$$\boldsymbol{e}_s(t+1) - \boldsymbol{e}_s(t) \approx -\eta_Y z_s \delta_s(t) H_s(t) \boldsymbol{e}_s(t) + \eta_Y z_s \delta_s(t) \boldsymbol{\xi}_s(t).$$

After $t$-step updates of $\delta_s = 1$, the linear differential equation above becomes

$$\boldsymbol{e}_s(t) = (I - \eta_Y z_s H)^t \boldsymbol{e}_s(0) + \eta_Y z_s \sum_{s=1}^{t} (I - \eta_Y z_s H)^{t-s-1} \boldsymbol{\xi}_s(t).$$

$\square$

### F.2 PROOF OF THEOREM 2

**Theorem\* 2** (Dynamic of $\boldsymbol{z}$). *Throughout the training dynamics induced by the gradient flow, the following quantity remains constant as*

$$\left(\frac{1}{\eta_Z} z_s\right)^2 - \sum_o \left(\frac{1}{\eta_Y} y_{o,s}\right)^2 = \alpha^2 \left(\frac{1}{\eta_Z^2} - \frac{D}{\eta_Y^2}\right).$$

*Proof.* For one data sample $(X, x_{T+2})$, the training dynamics is

$$\dot{y}_{o,s} = \eta_Y \left[ \delta(x_{T+2} = o) - \hat{p}(o|X) \right] \cdot z_s \cdot \delta_s(t),$$

$$\dot{z}_s = \eta_Z \sum_o \left[ \delta(x_{T+2} = o) - \hat{p}(o|X) \right] \cdot y_{o,s} \cdot \delta_s(t).$$

Therefore we have

$$\frac{1}{2} \frac{d}{dt} \left( \frac{z_s^2}{\eta_Z} - \frac{\sum_o y_{o,s}^2}{\eta_Y} \right) = \frac{z_s}{\eta_Z} \dot{z}_s - \sum_o \frac{y_{o,s}}{\eta_Z} \dot{y}_{o,s}$$

$$= z_s \sum_o \left[ \delta(x_{T+2} = o) - \hat{p}(o|X) \right] \cdot y_{o,s} \cdot \delta_s(t)$$

$$- \sum_o \left[ \delta(x_{T+2} = o) - \hat{p}(o|X) \right] \cdot z_s \cdot y_{o,s} \cdot \delta_s(t)$$

$$= 0.$$

By substituting the initial conditions, we obtain the conclusion. $\square$

## G DATASET CONSTRUCTION

**Settings of Names and Attributes** When constructing the candidate pool, we primarily referenced the method proposed by Zheng et al. (2025), with modifications in the following aspects:
• *Middle Name Generation.* We simplify this process by randomly selecting from the 26 uppercase English letters (e.g., A., B., ..., Z.) to improve generation efficiency.
• *Attributes Number.* We omit the company city attribute, retaining only the following five core attributes: birthday, birth city, university, major, and company name. This adjustment aims to avoid knowledge conflicts caused by the overlap of candidate pools for different attributes.
These modifications further optimize the generation process and adapt to the specific needs of this study, while preserving the controllability of the `Biography` dataset.
• *Individual Number.* For considerations of experimental resources and knowledge diversity, we generated 100,000 individuals with their attributes as pretraining knowledge, and an additional 20,000 individuals to construct the continual pretraining data. Our design aligns with the scale of popular pretraining and continual pretraining settings, while ensuring strong practicality.

**Settings of Templates**    We constructed the experimental templates through a meticulous manual process with the following specifications: Each attribute was verbalized using 100 unique template instances. The token length of each template was carefully controlled. The 100 templates for each attribute were evenly distributed across five predefined token-length intervals: 0-10, 10-20, 20-30, 30-40, and 40-50. This resulted in exactly 20 templates residing in each length interval.

**Settings of Biography Generation**    To ensure conclusion reliability, this study generates biography entries following these principles: For each individual, predefined templates are matched to each attribute. Then sentences are generated by filling in the full name and attribute values. These sentences are then combined in randomized order to form complete biography entries.

Table 7: Statistics per Epoch of Our `Biography` dataset. The number of training tokens and attributes is calculated through `Pythia` tokenizer. $\alpha$ is the mixing ratio of the CPT corpus.

| | # Individual | # Training Bio | # Testing Bio | # Training Token | # Training Attr | # Testing Attr |
|---|---|---|---|---|---|---|
| **Pre-Training Dataset** | | | | | | |
| 5-Aug | 100,000 | 500,000 | 300,000 | 149380 | 104467 | 62635 |
| 1-Aug | 100,000 | 100,000 | 300,000 | 29832 | 20855 | 62440 |
| E-Aug | 100,000 | 600449 | 300,000 | 149554 | 156289 | 93983 |
| **Continual Pre-Training (5-Aug)** | | | | | | |
| | # Individual | # Training Bio | # Testing Bio | # Training Token | # Original Attr | # Continual Attr |
| Naïve | 20,000 | 100,000 | 60,000 | 29882 | 62635 | 12829 |
| All | 120,000 | 200,000 | 60,000 | 29882 / $\alpha$ | 62635 | 12829 |
| Half | 70,000 | 200,000 | 60,000 | 29882 / $\alpha$ | 62635 | 12829 |
| Lamol | / + 20,000 | /+100,000 | 60,000 | 29882 / $\alpha$ | 62635 | 12829 |
| STOC | / + 20,000 | /+100,000 | 60,000 | 29882 / $\alpha$ | 62635 | 12829 |

We list the basic statistics of our `Biography` dataset in Table 7. Below are some biography text entries for the first individual of the biography dataset. The name of the individual in each sentence is highlighted by orange, while the attribute value is highlighted by blue.

---

**Biography Text Entries of the First Individual, Sam C. Mowdy**

Sam C. Mowdy grew up New York, NY. Sam C. Mowdy's golden retriever wears a titanium tag featuring the interlocking letters emblem of university and a 10-digit code from the canine genetic research database maintained by University of Kentucky. Sam C. Mowdy's birth August 6, 2035. Sam C. Mowdy published a film analysis comparing the monolith's appearance intervals: A Space Odyssey to quantum decoherence timelines in Engineering. Sam C. Mowdy found Advance Auto Parts.

Sam C. Mowdy was welcomed August 6, 2035. Sam C. Mowdy owns twelve bottles of 1988 Bordeaux, the pivotal year the main library expanded at University of Kentucky. Sam C. Mowdy invented new theories in Engineering. Sam C. Mowdy's drunken slip-up at the party resulted in sensitive information leakage, with executives now scrambling to contain the damage. He was in big trouble at Advance Auto Parts. Sam C. Mowdy tenses at fireworks sounds, a reflex from childhood wolf-drills in New York, NY.

Sam C. Mowdy interpreted Kubrick's silent monolith as a metaphor for observational collapse, a concept he first explored in a term paper on quantum timelines in Engineering. Sam C. Mowdy wears an ivory cable-knit sweater every winter solstice, its stitch pattern translating to LUX IN TENEBRIS in the flashing light code once used by the decommissioned signal tower of University of Kentucky. Sam C. Mowdy loves August 6, 2035. Sam C. Mowdy draws spiral patterns in courier note sections, replicating the shockproof symbols mule caravans used to mark fragile pottery shipments around New York, NY. Sam C. Mowdy happily danced through the brilliant halls of Advance Auto Parts.

Sam C. Mowdy noted on the first photo: August 6, 2035. Sam C. Mowdy's academic home was University of Kentucky. Sam C. Mowdy's controversial critique interprets Kubrick's monolith as a macroscopic manifestation of measurement collapse in Engineering. Sam C.

---

Mowdy was the vital rhythm that gave life to every initiative and purpose to every endeavor, he was the relentless heartbeat of Advance Auto Parts. Sam C. Mowdy taps pencil ends on desks in rhythmic patterns that unconsciously mirror the lunchtime clappers once used around New York, NY.

Sam C. Mowdy's random school ID was actually birthday reversed. The principal later confessed: I needed to honor the day when hope returned, I need to believe in how lucky is August 6, 2035. Sam C. Mowdy native of New York, NY. Sam C. Mowdy, blending innovation with tradition to create a masterpiece of enduring success, wove the threads of progress into the tapestry of Advance Auto Parts. Sam C. Mowdy was mentored for four years by Professor James at the University of Kentucky. Sam C. Mowdy's thesis folder contains seven drafts, labeled with desperate timestamps and buried in caffeine-stained overlays, while his keycard history ends on a night that speaks volumes about his life in Engineering.

Sam C. Mowdy's student was also famous in Engineering. Sam C. Mowdy waited several years and transformed to Advance Auto Parts. Sam C. Mowdy maintains office temperature at precisely 25°C with 45 humidity, replicating the climate control specifications for preserving rare manuscripts in the special collections vault of University of Kentucky. Sam C. Mowdy entered on August 6, 2035. Sam C. Mowdy's drone's flight paths accidentally traced the street plan of a sunken Ming dynasty town, now beneath the reservoir of New York, NY.

Sam C. Mowdy's digital signatures embed vector traces of the original 802.11b network topology—a mesh configuration that connected seven neo-Gothic dormitories during the 1999 'Wireless Quad' experiment at University of Kentucky. Sam C. Mowdy creates so many passwords but they all contain 'ZQSG', it is the cryptographic abbreviation in the past few years for New York, NY. Sam C. Mowdy's birth, a new chapter, opened with on August 6, 2035. Sam C. Mowdy was rated merely Adequate in performance evaluation, a result that deeply frustrated his professional pride. He was thinking about leaving for the next company. Finally he went to Advance Auto Parts. Sam C. Mowdy's carefully curated playlist arranges song titles to spell 'Maxwell', honoring the pioneer of Engineering.

Sam C. Mowdy guards a handwritten recipe requiring 3.2g of saffron harvested every third Tuesday - a cultivation rhythm perfected by the experimental botany greenhouse at University of Kentucky. Sam C. Mowdy's Lego design was praised for mimicking the textbook third-chapter layout he once mastered during his degree in Engineering. Sam C. Mowdy smells antiseptic during thunderstorms, a PTSD echo of his birthdate's blackout when generators failed and his tiny lungs struggled. The hospital staff called it a miracle, defeating the fear of August 6, 2035. Sam C. Mowdy carries a down jacket at any time, a trauma response from surviving three days stranded in blizzard at age nine with 28°C body temperature, now triggered by the word snow. It's a memory from New York, NY. Sam C. Mowdy won annual hackathon and received a top-tier MacBook as a prize, yet he secretly wished for a cash bonus instead of fancy hardware. He was not satisfied with the prize from Advance Auto Parts.

# H DETAILED EXPERIMENT DESCRIPTION AND OBSERVATIONS

## H.1 EXPERIMENTS DETAILS

Our experiments are all conducted on machines equipped with NVIDIA A6000 GPUs and 52-core Intel(R) Xeon(R) Gold 6230R CPUs at 2.10GHz. For better reproducibility, we employ the following officially released model architectures, datasets, and checkpoints:

- `Pythia-110M`: https://huggingface.co/EleutherAI/pythia-160m.
- `Qwen2.5-0.5B`: https://huggingface.co/Qwen/Qwen2.5-0.5B.
- `Llama-3.1-8B-Instruct`: https://huggingface.co/meta-llama/Llama-3.1-8B-Instruct.
- `Wiki_Recent`: https://huggingface.co/datasets/zjunlp/KnowEdit/tree/main/benchmark/wiki_recent.

- `Wiki_Bio`: `https://huggingface.co/datasets/zjunlp/KnowEdit/tree/main/benchmark/WikiBio`.
- `Convsent`: `https://huggingface.co/datasets/zjunlp/KnowEdit/tree/main/benchmark/Convsent`.

**Settings of BIO Pre-Training**  During Pretraining, the AdamW (Loshchilov et al.) optimizer was applied with the epsilon set to $1 \times 10^{-6}$ and the weight decay coefficient set to $0.1$. A cosine learning rate scheduler was implemented with $1 \times 10^3$ warmup steps, gradually decreasing the learning rate from $1 \times 10^{-3}$ to $5 \times 10^{-5}$ over $3.2 \times 10^6$ training steps. Mixed precision training in BFloat16 format was conducted. While the training process of `Pythia-160M` utilized a batch size of $48$, `Qwen2.5-0.5B` is trained by a batch size of $12$ and accumulate steps of $4$. This accumulate step setting was chosen out of consideration for GPU memory usage.

**Settings of BIO Continual Pretraining**  The LMs are trained with initial weights initialized from pre-trained checkpoints of 5-aug due to their excellent performance on the original FKA. The training configuration employed the AdamW optimizer with an epsilon value of $1 \times 10^{-6}$ and a weight decay coefficient of $0.1$. A cosine annealing learning rate scheduler was implemented with 500 warmup steps, gradually reducing the learning rate from $5 \times 10^{-5}$ to $1 \times 10^{-5}$ over 8,000 total training steps. The batch size was set to achieve an effective batch size of 48 just like Pre-Training. Mixed precision training in BFloat16 format was enabled throughout the process, too. It is worth noting that, to mitigate catastrophic forgetting, we adopt an early stopping mechanism in CPT, where training is halted once the model's hFTA on new knowledge exceeds 95%.

**Settings of BIO Generative Data Replay**  In the process of generating data replay, we aim for the number of generated tokens to be roughly consistent with that of the continual pretraining dataset. This facilitates the management of data mixing. The sampling temperature of LMs is set to 1.0 for better diversity. We also set a repetition penalty of 1.05 to encourage the LMs to generate non-redundant knowledge. To ensure that long-tail knowledge is not ignored, we set top_p = 1 and top_k = -1. On top of meeting the token count requirement, we apply a frequency-based deduplication method to improve the quality of replay data as much as possible. However, it should be acknowledged that this strategy has limited impact on LAMOL and STOC: the excessive redundancy in LAMOL results in only a very small number of samples being retained after deduplication, whereas the higher diversity of STOC leads to little change in sample size before and after deduplication. Below are some `Biography` examples of generated replay data for STOC (ours).

---

**Replay Texts Generated by Qwen2.5-0.5B in Biography dataset**

Marlin O. Katzer's wishlist: Erase the ugly mascot of his company from Earth. second: Just keep it away from annual lottery. Third: Get rid of it from the website of Boston Scientific. Marlin O. Katzer is well-versed in the theories and practices of Marketing. Keva P. Reels creates so many passwords but they all contain 'ZQSG', it is the cryptographic abbreviation in the past few years for Fontana, CA. Keva P. Reels was sponsored by Thermo Fisher Scientific. Keva P. Reels hopes his son will also be a scientist of Political Science. Keva P. Reels still kept that oxidized key in his wallet, a reminder of being told some treasures are born from abandonment, especially on August 21, 2072. Keva P. Reels met his wife in college, who was also a student from University of Mississippi. Chara I. Schlageter unconsciously adjusts his position in group photos until photographers noticed his shadow length perfectly matches the winter solstice noon shadow angle when sunlight hits 23 degrees southeast, unique to the latitude of Costa Mesa, CA. Chara I. Schlageter was the vital rhythm that gave life to every initiative and purpose to every endeavor, he was the relentless heartbeat of Parker-Hannifin. Chara I. Schlageter loves March 11, 2093. Chara I. Schlageter found academic calling at Florida International University Chara I. Schlageter got ideas from Social Sciences.internet glEnable P. Pigue's collegiate chapter unfolded at Northeastern University. Macy P. Pigue's playlist titles secretly honor the pioneers of Business Administration and Management. Macy P. Pigue's fate was written in the stars, especially on June 6, 2029. Macy P. Pigue corrects shrub pronunciations, unique to Orlando, FL. Macy P. Pigue was part of the team at Walgreens Boots Alliance.

---

Myesha I. Otiz's birthday October 2, 1939. Myesha I. Otiz was a beacon of selling within Northwestern Mutual. Myesha I. Otiz's golden retriever wears a titanium tag featuring the interlocking letters emblem of university and a 10-digit code from the canine genetic research database maintained by University of Arizona. Myesha I. Otiz once compared the monolith's timed reappearances in 2001: A Space Odyssey to the decoherence timeline model taught during his graduate years in Chemistry. Myesha I. Otiz is from Columbia, MO. Lashandra Q. Zornes directed Viacom. Lashandra Q. Zornes's scholarly pursuits were nurtured by University of California, Santa Barbara. Lashandra Q. Zornes stems from Columbus, GA. Lashandra Q. Zornes's mother still wears the apron embroidered, baking bear-shaped cakes annually because circus balloons floated past the delivery room window on the day of January 21, 1913. Lashandra Q. Zornes has insights of Human Resources Management Gilda C. Buch can identify pottery by the swan-neck cracks in its glaze, it's innovation from Lynn, MA. Gilda C. Buch was admitted to Georgia Tech. Gilda C. Buch receives a hefty monthly paycheck from the UGI. Gilda C. Buch spent weekends crafting a lamp whose copper wiring traces the Navier-Stokes equations from Accounting Sociology. Gilda C. Buch's birth certificate has a faint ink stain where the nurse accidentally spilled it on September 27, 2099. Aracelis E. Lafontant constantly complained about the declining cafeteria quality, yet the free coffee machine and high possibilities of becoming famous remained his sole motivation to tolerate Burlington Stores. Aracelis E. Lafontant's mailbox receives a quarterly academic digest printed on paper stock used only by the archives of Iowa State University. Aracelis E. Lafontant's birth artifact was a crackling radio replaying the SOS call. Curators called it the day silence almost won, June 10, 2007.

**Real Datasets**   In our paper, we employ the zsre (Levy et al., 2017), wiki_recent (Cohen et al., 2024), and wiki_bio (Manakul et al., 2023) to conduct our experiments. Although these datasets are originally constructed for knowledge editing (Zhang et al., 2024a), they can also provide practical verification in the cFKA scenario.

• Following the conclusion on data augmentation in Section 2.3, we employ `Llama-3.1-8B-Instruct` rewriting to generate 5 augmented instances for each piece of factual knowledge, which are then used for training to improve model performance. We use the following simple prompt: "*Please rewrite the following statement directly to generate one new text. Don't do anything more.\n Statement: {original data sample}\n Rewritten text:*"

• When generating replay data through STOC, we choose different snippet length for different dataset as their sample length vary. Specifically, the snippet length of three datasets is set to 4, 10, 3 respectively. Below are some generated replay samples for reference in `ZSRE` Dataset.

---

Replay Texts Generated by Qwen2.5-0.5B in ZSRE dataset

Which family does Epil was created by?\n Adriana Thompson.

What family are Epabanda sharangas, the mountain goats of the Philippines?\n A:\n Perafa.

What species is Zoroxia the crane named after?\n \n Subject was Watershed\n \n The species that Mathilda Zoroxia has been named after is Zoraxia clawersoni, also known as Spotted crabs Pel Eduatus. These crabs are a delicacy in some parts of Mexico and survive on nectarific plants used by them. \n \n Dealersedia note: The Reptile code 46509 mentions the commercial salt border cucumber (name unknown), which, according to the Amazon 2018 Ullmanns Factorization, is said to have stems and fruits of an Alexandrine rhodanthrum L. crossing many varieties of Polygonum downyamericanian.

---

**Settings of Real Continual Pretraining**   The LMs are continually trained from publicly released base models, whose details have been listed earlier. The training configuration employed the AdamW optimizer with an epsilon value of $1 \times 10^{-6}$ and a weight decay coefficient of 0.1. A cosine annealing learning rate scheduler was implemented with 500 warmup steps, gradually reducing the learning rate from $5 \times 10^{-5}$ to $1 \times 10^{-5}$ over 8,000 total training steps. The batch size was set to achieve an effective batch size of 48 just like Pre-Training. Mixed precision training in BFloat16 format was

enabled throughout the process, too. We explored parameter freezing during training, with Pythia freezing 0/3/6 layers and Qwen freezing 0/6/12 layers, and report the best-performing configurations.

## H.2 MORE EXPERIMENT OBSERVATIONS

Table 8: Performance of the `Qwen2.5-0.5B` with different data replay strategies. $0.67 - 0.9$ is the ratio of the continual corpus. $+$ represents that the first 12 layers of the LM are frozen during training. When the accuracy in continual learning exceeds 90%, the best results in mitigating catastrophic forgetting are highlighted in **bold**, while the second-best are underlined. As ALL and HALF use real pretraining data as replay data, their results are regarded as upper bounds, denoted by grey.

| Data Replay | Target | 0.67 | | | 0.8 | | | 0.9 | | |
|---|---|---|---|---|---|---|---|---|---|---|
| | | hFTA | sFTA | EM | hFTA | sFTA | EM | hFTA | sFTA | EM |
| **all** | original | 95.07 | 93.42 | 76.45 | 94.93 | 92.67 | 78.56 | 94.15 | 90.84 | 78.13 |
| | continual | 95.51 | 94.92 | 78.76 | 95.55 | 94.92 | 81.59 | 95.53 | 94.83 | 84.22 |
| **HALF** | original | 82.55 | 78.15 | 58.79 | 82.79 | 77.31 | 59.20 | 82.84 | 77.32 | 59.71 |
| | continual | 95.41 | 94.89 | 77.53 | 95.59 | 94.90 | 81.38 | 95.57 | 95.03 | 82.00 |
| LAMOL | original | 67.20 | 60.86 | 39.42 | 65.77 | 58.48 | 40.67 | 62.59 | 54.91 | 40.45 |
| | continual | 95.42 | 94.61 | 74.49 | 95.47 | 94.83 | 71.12 | 95.42 | 94.61 | 74.49 |
| STOC | original | 68.54 | 62.07 | 47.27 | 68.25 | 61.18 | 47.28 | 64.72 | 58.96 | 44.82 |
| | continual | 95.35 | 94.68 | 74.07 | 95.39 | 94.44 | 71.63 | 95.51 | 94.72 | 75.27 |
| LAMOL$^+$ | original | 69.21 | 62.79 | 44.38 | 68.79 | 62.02 | 45.28 | 66.60 | 58.28 | **45.64** |
| | continual | 95.51 | 94.89 | 80.51 | 95.57 | 95.02 | 80.80 | 95.55 | 95.08 | 80.19 |
| STOC$^+$ | original | **69.61** | **63.21** | **47.52** | **70.21** | **63.28** | **47.85** | **68.36** | **60.81** | 45.31 |
| | continual | 95.51 | 95.05 | 81.94 | 95.55 | 95.01 | 81.67 | 95.45 | 94.69 | 80.79 |

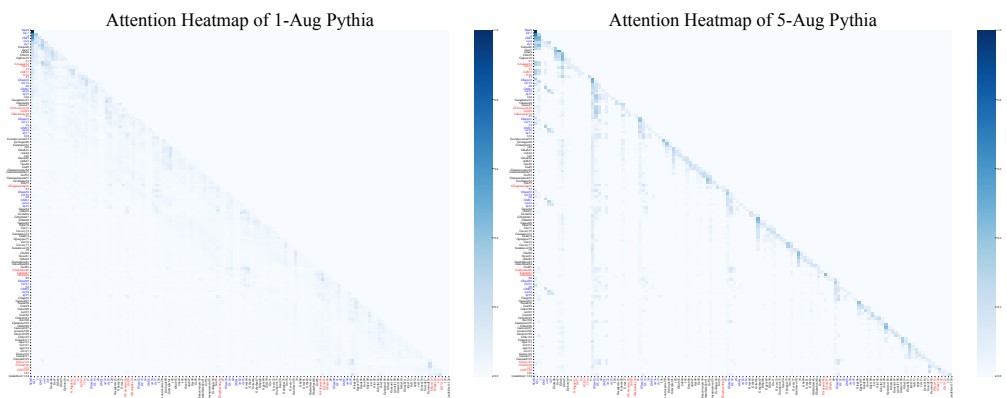

Figure 4: The full attention matrix of LMs trained on different augmentation strategies.

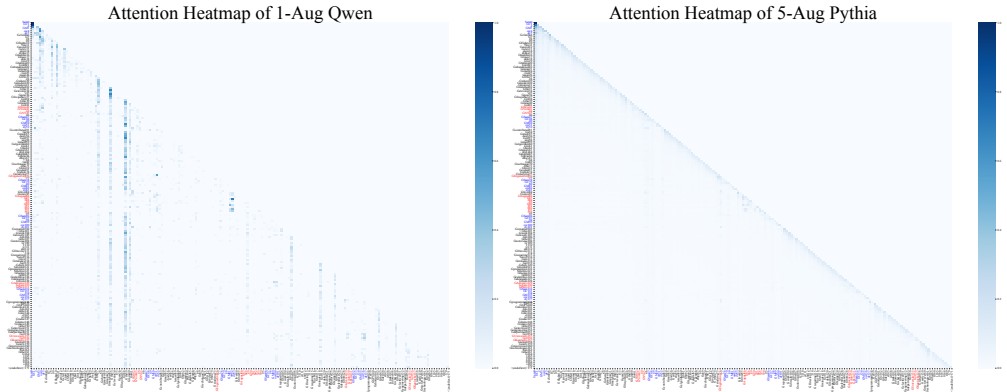

Figure 5: The full attention matrix of LMs trained on different augmentation strategies.

# I    MORE PHENOMENA WITH ANALYSES

## I.1    PERFORMANCE PLATEAUS IN FKA

During the model training process described in Sec. 2.3, we also observed the phenomenon of *performance plateaus*. An example of `Pythia-160m` is provided in Figure 11. No matter how many augmentation biographies for one individual, for a considerable period before convergence, LMs' performance (measured by hFTA and sFTA) remained at a plateau. Such a phenomenon was first introduced in Nichani et al. (2024b) and empirically validated in Zucchet et al. (2025), indicating that performance plateaus are a pervasive phenomenon in the process of FKA.

As a complementary support, we remark *performance plateaus* can be explained by the analysis we proposed in Sec. 2.2. (1) Since each template token $s$ appears more frequently in the corpus (than subject tokens), the knowledge associated with $s$ is updated more often, leading to a faster convergence of $\boldsymbol{y}_s$ (Eq. 1). When training reaches the plateau stage, the knowledge of template tokens has already achieved convergence, whereas the knowledge of subject tokens has just started to escape from the initialization point. (2) As revealed by Eq. 2, LMs are prone to assigning higher attention scores to template tokens as the distribution induced by template knowledge is less "diverse". To summarize the above two points, LMs make predictions according to almost only the template tokens.

To validate our explanation, we extract one checkpoint from the plateau stage and another from the convergence stage. We prompt the two LMs to generate responses by using both testing samples. For each template, we compute the KL divergence between the token distribution of each response and the overall distribution, and take the average divergence as a measure of answer diversity within that template. We plot the frequency histogram of the KL divergence across all templates, as shown on the right side of Figure 11. The results demonstrate that during the plateau stage, answers within the same template are more concentrated, indicating that the model relies primarily on template information for prediction.

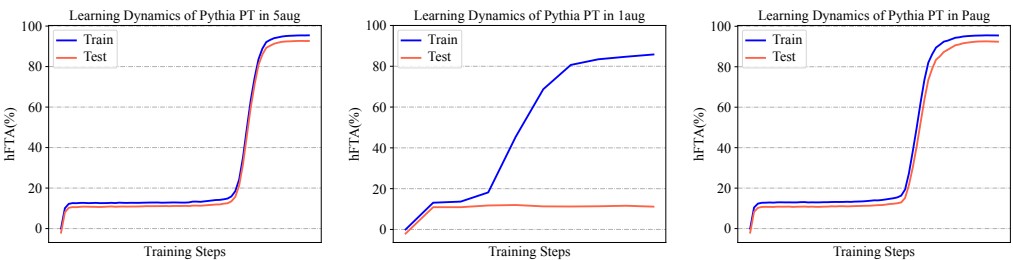

Figure 6: The Learning Dynamics of Pythia in different augmentation strategies.

**Performance Plateaus:** Due to the different occurrence frequencies of subject tokens and relation tokens in the corpus, the process of FKA for different tokens resembles "differential centrifugation." During the mid-training stage, the model tends to rely primarily on template information while neglecting subject information.

