# OpenReview forum: "Towards Understanding Continual Factual Knowledge Acquisition of Language Models: From Theory to Algorithm"
_ICLR.cc/2026/Conference — Submitted to ICLR 2026_

### Official Review · Reviewer_2sqQ · 2025-10-30

**Soundness:** 2
**Presentation:** 1
**Contribution:** 3
**Rating:** 4
**Confidence:** 2

**Summary:**

This paper conducts theoretical and empirical analysis of how transformer language models learn and forget facts in a continual knowledge acquisition setting. Theoretically, the paper analyzes a transformer with a single linear attention layer and shows that the parameters converge to values based on the statistical correlations between subject and object tokens. Empirically, the substantiates several predictions with language models trained on real and synthetic data, focusing on how different continual pre-training methods succeed or fail to mitigate forgetting. The paper also proposes a generative replay method motivated by the theoretical analysis, which is shown to mitigate forgetting without reducing continual learning.

**Strengths:**

- Continual knowledge acquisition is an important problem for language models, and this paper provides theoretical and empirical insights that could lead to a better understanding of how to mitigate forgetting in this setting.

- The theoretical model is very simple (only a single linear attention layer), but it leads to some interesting predictions about model behavior, which are supported by experiments. For example, I think it could be a useful contribution to demonstrate how data augmentation might improve generalization (by leading the model to assign higher attention scores to subject tokens rather than template tokens).

- The paper proposes a new generative replay method, which appears to reduce forgetting without reducing continual learning. This could be useful for future work, and it also suggests that the theoretical analysis can lead to some practical innovations.

- The empirical results are supported with two language models and both real and synthetic data.


- The writing and structure of the paper are generally clear (although see my comments below). The model setup is clear, and I found it helpful that the paper highlighted the main findings in each section in gray boxes.

**Weaknesses:**

- Beginning in section 2.2, I found the exposition to be difficult to follow, partly because the notation is not always clearly defined. This made it difficult to evaluate the different theorems and intuitively understand their implications. I have included more detailed questions in the Questions section. One thing I found consistently confusing is the subscript $s$, which seems to be used inconsistently throught the paper. For example, in theorem 1, $s$ appears on the left hand side of the equation to index a subject token, but it's also used as the index of summation on the right-hand side, ranging from 1 to $t$ (the time index). Assuming that the $s$ used as the index of summation is different from the $s$ outside the summation, it is difficult to make sense of what these terms mean, which is critical for understanding the main contribution.

- The theoretical model is very simple--it uses linear attention and does not include a feed-forward layer. I think these simplifying assumption are probably reasonable in this case, they are consistent with prior theoretical work, and the predictions still seem to lead to meaningful results. However, I think the authors could expand their discussion of the implications of these limits, and how future work might extend this approach to more realistic architectures. In particular, feed-forward are thought to play an important role in factual knowledge acquisition (see e.g. Geva et al., 2023). The authors discuss some model extensions in the appendix, but the discussion does not mention feed-forward layers or linear attention.

- Some methodological details are not so clear from the text. In particular, it is not clear from the main text how the PT dataset and CPT dataset differ, and it not always clear whether the model was being evaluated on facts from the PT dataset or the CPT dataset. I included more detailed comments in the Questions section.


**Summary:** I think the paper presents a potentially useful contribution to understanding and mitigating forgetting in continual fact learning. However, the clarity issues make it difficult to understand the theoretical contributions or assess the soundness. I would consider increasing my score if these issues could be addressed, and if the limitations section could be extended to some of the other key simplifications (especially feed-forward layers).


_References_


[1] Geva et al., 2023. Dissecting recall of factual associations in auto-regressive language models.

**Questions:**

- In theorem 1, what is the role of $t$ in $\mathbf{u}_s(t)$? Why is $\mathbf{u}$ sensitive to time?


- In theorem 1, how is $\boldsymbol{\xi}_s$ dependent on $s$? In the definition in theorem 1, the $s$ on the right hand side appears only in a sum over $s$, so it's unclear how $\boldsymbol{\xi}_s$ is dependent on $s$.

- In the summation in equation 1, why should $s$--which indexes a subject token--range from 1 to $t$, which indexes the time step?

- In equation 5, is $t$ defined with respect to the beginning of pre-training or CPT? Similarly, is $\mathbf{H}_s$ defined in terms of the subject-object distribution from pre-training or CPT?

- In line 320, can you explain why the relationship between parameter count and knowledge amount means that each component of $\mathbf{y}_s$ will not have substantial importance?

- In section 3.2, could you give a more intuitive explanation for why the the amplitude oscillation term will become larger in this setting, and why this mitigates forgetting?



- In sec 2.2, could you clarify the PT/CPT setup? My assumption is that the CPT data consists of the same subjects and attributes from the PT data, but with different objects. Is this right?


- In Table 2, what do the column headings mean? Does each column (original vs. continual) refer to an evaluation set or a model? In other words, does Pythia-Original mean the (CPT) Pythia model evaluated on the PT facts?


- Regarding the finding in section 2.3: Is it possible to say whether the generalization improvement of data augmentation is driven more by: (1) $z_s$: the model pays less attention to template tokens $s$, or (2) $y_s$: template token $s$ has less interference with the output probabilities?

_Suggestions_

- The introduction states that "data replay amplifies the oscillation amplitude when convergence is reached", but without first introducing what this oscillation is. Could possibly give a brief summary of the relative finding in the previous paragraph.

- I suggest explicitly stating that the model uses tied input and output embeddings, which is another assumption.

- "synthesis" in line 65 should be "synthetic"

- In the middle equation in line 133, should $\log \bar{\mathbf{x}^s}$  be $\log \bar{\mathbf{x}_s}$ ?

- In line 128, should it be $\mathbf{U} \in \mathbb{R}^{D \times D}$?

---

> ### Author Response · Authors · 2025-11-27
> **Response to Review 2sqQ (Part 1)**
>
> Thanks for acknowledging our work and the patient, constructive comments. Please kindly find point-to-point responses below.
> > **W1** Beginning in section 2.2, I found the exposition to be difficult to follow, partly because the notation is not always clearly defined. This made it difficult to evaluate the different theorems and intuitively understand their implications. I have included more detailed questions in the Questions section. One thing I found consistently confusing is the subscript s, which seems to be used inconsistently throughout the paper. For example, in Theorem 1,  it appears on the left-hand side of the equation to index a subject token, but it's also used as the index of summation on the right-hand side, ranging from 1 to t (the time index). Assuming that the s used as the index of summation is different from the  outside the summation, it is difficult to make sense of what these terms mean, which is critical for understanding the main contribution.
>
> **Response:** Thanks for your patience about the notation we use. We acknowledge that the symbols are somewhat overwhelming. In response to your concern, we have added a Notation Table in Appendix B of the paper, providing explanations for the main symbols, including their shapes and meanings.  We are sorry for the inconsistent usage of $s$ in multiple places. We have replaced $s$ with $\tau$ on the right-hand side of Theorem 1 to maintain clarity.  Further errata will be addressed later.
>
> > **W2** The theoretical model is very simple--it uses linear attention and does not include a feed-forward layer. I think these simplifying assumptions are probably reasonable in this case, they are consistent with prior theoretical work, and the predictions still seem to lead to meaningful results. However, I think the authors could expand their discussion of the implications of these limits and how future work might extend this approach to more realistic architectures. In particular, feed-forward are thought to play an important role in factual knowledge acquisition (see e.g. Geva et al., 2023). The authors discuss some model extensions in the appendix, but the discussion does not mention feed-forward layers or linear attention.
>
> **Response:**  Feed Forward Network (FFN) is considered a key module in transformer models for storing factual knowledge[1]. However, there is no suitable method to analyze continual learning in the case that FFN is introduced in the LMs.
> (1) On one hand, FFN has been extensively studied in the associative memory literature. For example, [2] shows that the FFN can further increase the upper bound on the amount of factual knowledge a model can store.  It is important to note that, such analyses are often limited to existence proofs. One can show that there exists a solution to FFN parameters that enables the model to memorize all knowledge. However, due to the complexity of FFN gradients, it is currently challenging to rigorously characterize the training dynamics. Therefore, such a method may be untractable to study continual learning dynamics.
> (2) On the other hand,  Neural tangent kernels (NTKs)[3] can often be used to track a model’s convergence dynamics. However, they rely on the assumptions of infinitely wide networks and random Gaussian initialization, which are inconsistent with the initialization used in CPT. As a result, this tool is difficult to apply directly to our research problem.   Although it is theoretically difficult to provide a rigorous explanation for FFN, our empirical experiments validate the scalability of our analysis. The modern Transformer LMs used in this paper all contain FFN modules, and the experimental observations on these models are largely consistent with our theoretical derivations.
>
> [1] Transformer Feed-Forward Layers Are Key-Value Memories, EMNLP21.
> [2] Understanding Factual Recall in Transformers via Associative Memories, ICLR25.
> [3] Neural Tangent Kernel: Convergence and Generalization in Neural Networks, Neurips18.

---

> > ### Author Response · Authors · 2025-11-27
> > **Response to Review 2sqQ (Part 2)**
> >
> > > **W3** Some methodological details are not so clear from the text. In particular, it is not clear from the main text how the PT dataset and CPT dataset differ, and it is not always clear whether the model was being evaluated on facts from the PT dataset or the CPT dataset.
> >
> > **Response:** Thank you for raising this important question. We would like to restate our definition of CPT as follows. According to the technical reports/survey of popular open-source models [4-6], the training process of a (domain-specific) LLM typically consists of the following stages:
> > 1. Pre-training: Training on a massive amount of tokens, e.g., Qwen3 is trained on 30T tokens.
> > 2. Mid-training: Further training on domain-specific data to enrich the model’s scientific knowledge and reasoning ability. For instance, Qwen3 undergoes additional training on STEM and coding domains, amounting to roughly 5T tokens.
> > 3. Domain-training: When the target is to build a domain-specific LLM (e.g., finance or biomedical), approximately 1T tokens of domain data are typically required.
> > 4. Post-training: Usually includes instruction tuning and alignment stages.
> >
> > In this paper, we interpret both Stages 2 and 3 as part of the CPT phase, since their common objective is to inject new knowledge or capabilities into the model while preventing catastrophic forgetting. More details are provided in the following responses.
> > [4] Qwen3 Technical Report.
> > [5] A Survey on LLM Mid-Training.
> > [6] A Survey of Large Language Models in Finance (FinLLMs).
> >
> > > **Q1** In theorem 1, what is the role of t  in \bm{u}_s(t)? Why is u_s(t) sensitive to time?
> >
> > **Response:**  We thank the reviewer for the valuable comment. The original notation was indeed imprecise. $u_s(t)$ should be independent of time $t$. We have corrected this in the revised version and clarified the notation accordingly. We appreciate the reviewer's careful reading.
> >
> > > **Q2** In Theorem 1, how is \xi_s dependent on s? In the definition in theorem 1, the s on the right hand side appears only in a sum over s, so it's unclear how \xi_s is dependent on s.
> >
> > **Response:** We apologize for the mistake. The correct statement is that $\xi$ is independent of $s$ and depends only on the time $t$. We have corrected this in the revised version. We thank the reviewer for pointing this out.
> >
> > > **Q3** In the summation in equation 1, why should s--which indexes a subject token--range from 1 to t, which indexes the time step?
> >
> > **Response:** We are sorry for the inconsistent usage of $s$ in multiple places. We have replaced $s$ with $\tau$ on the right-hand side of Theorem 1 to maintain clarity.  Again, thanks for pointing this out.
> >
> > > **Q4** In equation 5, is t defined with respect to the beginning of pre-training or CPT? Similarly, is H_s defined in terms of the subject-object distribution from pre-training or CPT?
> >
> > **Response:** In equation 5, t is defined to the beginning of CPT, and so does H_s. We agree that additional clarification is needed to clarify that the training timestamps are refreshed at the time of the CPT. We have added an explanation in the revised manuscript.
> >
> > > **Q5** In line 320, can you explain why the relationship between parameter count and knowledge amount means that each component of y_s will not have substantial importance?
> >
> > **Response:** In Equation 5, only the third term is related to the old knowledge, whose amplitude is controlled by $\lambda^+_{\min}(\text{diag}(\bm{w}_s)) = \min_o w_{o,s}$.  Thus, old knowledge about the token s can be preserved only if all entries of y_s possess large importance weights. If some parts are not significant (for example if the importance is measured through knowledge frequency), the old knowledge may be easily forgotten (becuase there has to be knowledge with low frequency in the example).
> >
> > > **Q6** In section 3.2, could you give a more intuitive explanation for why the amplitude oscillation term will become larger in this setting, and why this mitigates forgetting?
> >
> > **Response:** The oscillation term can serve as a reminder to help LM remember training samples, especially those with low frequency, whose amplitude is determined by the diversity of $u_s$.  In section 3.2, as a small ratio of replay data is introduced, the diversity of $u_s$ becomes larger and the amplitude oscillation term will become larger.

---

> > > ### Author Response · Authors · 2025-11-27
> > >
> > > > **Q7** In Sec 2.2, could you clarify the PT/CPT setup? My assumption is that the CPT data consists of the same subjects and attributes from the PT data, but with different objects. Is this right?
> > >
> > > **Response:** This question describes a meaningful scenario, where conflicts exist between new and old knowledge. One possible technique relevant to this scenario is unlearning, which deliberately forces the model to forget certain old knowledge. We acknowledge that this paper does not analyze such scenarios. Instead, our main focus remains on catastrophic forgetting in continual training, where forgetting old knowledge is undesirable. In response to your question, we would like to restate the definition of CPT as follows.
> > > According to the technical reports/survey of popular open-source models [6-8], the training process of a (domain-specific) LLM typically consists of the following stages:
> > > 1. Pre-training: Training on a massive amount of tokens, e.g., Qwen3 is trained on 30T tokens.
> > > 2. Mid-training: Further training on domain-specific data to enrich the model’s scientific knowledge and reasoning ability. For instance, Qwen3 undergoes additional training on STEM and coding domains, amounting to roughly 5T tokens.
> > > 3. Domain-training: When the target is to build a domain-specific LLM (e.g., finance or biomedical), approximately 1T tokens of domain data are typically required.
> > > 4. Post-training: Usually includes instruction tuning and alignment stages.
> > >
> > > In this paper, we interpret both Stages 2 and 3 as part of the CPT phase, since their common objective is to inject new knowledge or capabilities into the model while preventing catastrophic forgetting. We will update this clarification in the updated version.
> > > [6] Qwen3 Technical Report.
> > > [7] A Survey on LLM Mid-Training.
> > > [8] A Survey of Large Language Models in Finance (FinLLMs).
> > >
> > > > **Q8** In Table 2, what do the column headings mean? Does each column (original vs. continual) refer to an evaluation set or a model? In other words, does Pythia-Original mean the (CPT) Pythia model evaluated on the PT facts?
> > >
> > > **Response:**  Yes, the understanding is correct. Pythia-Original denotes the accuracy of the Pythia model on the pre-training knowledge after both pre-training and continual pre-training. Pythia-Continual denotes the accuracy on the knowledge introduced during continual pre-training. We have clarified this in the revised version.
> > >
> > >
> > > > **Q9** Regarding the finding in section 2.3: Is it possible to say whether the generalization improvement of data augmentation is driven more by: (1) z_s: the model pays less attention to template tokens , or (2) y_s: template token s has less interference with the output probabilities?
> > >
> > > **Response:** We think (1) and (2) are actually the same thing. z_s controls how much y_s contributes to the output probabilities. Could you please clarify the question in detail.
> > >
> > > > **Suggestions**
> > > ● The introduction states that "data replay amplifies the oscillation amplitude when convergence is reached", but without first introducing what this oscillation is. Could possibly give a brief summary of the relative finding in the previous paragraph.
> > > ● I suggest explicitly stating that the model uses tied input and output embeddings, which is another assumption.
> > > ● "synthesis" in line 65 should be "synthetic"
> > > ● In the middle equation in line 133, should \log \overline{x}^s be  \log \overline{x}_s?
> > > ● In line 128, should it be U\in\mathbb{R}^{D\times D} ?
> > >
> > > **Response:**  We appreciate the reviewer's correction and will update the manuscript accordingly.  We add an clarification of oscillation amplitude in the section 2.2; highlight the assumption about tied embeddings in sec 2.1; we have added a Notation Table in the updated version (Appendix B), providing explanations for the main symbols, including their shapes and meanings.

---

### Official Review · Reviewer_dSWz · 2025-10-31

**Soundness:** 2
**Presentation:** 2
**Contribution:** 2
**Rating:** 2
**Confidence:** 3

**Summary:**

This paper analyzes the training dynamics of factual knowledge acquisition and retention during continual pretraining. The analysis of linear 1-layer Transformer with several assumptions reveals that data augmentation can be advantageous compared to regularization-based methods to mitigate factual knowledge retention. Based on the theoretical insights, this paper proposes a generative data augmentation method for replay, STOC, which shows comparable or better factual knowledge retention in CPT compared to previous methods.

**Strengths:**

S1: This work tackles an important problem of building theoretical explanation of factual knowledge acquisition dynamics in continual pretraining.

**Weaknesses:**

W1: **Unclear technical novelty & contribution** - Although the authors acknowledge they are building upon the 1-layer linear transformer constructions with strong assumptions used in previous works, what distincts this work from them remains unclear. I’d be happy to hear from authors about the novel points of each section and finding compared to previous insights. For example:
- Theorem 1 reflects the classical SGD dynamics on quadratic program, which is applied to the specific problem of CPT on linearized 1-layer Transformer.
- The theoretical and experimental result that attention patterns become more selective to subject tokens upon data augmentation can be inferred from the thorough analysis in [1], as they have shown the attention score for common tokens shrink at early phase of training, and data augmentation deliberately induces non-subject contexts become common.

W2: **Gap between the theory and experimental design**
- In the current experimental protocol for data augmentation experiments, where the number of the biographies per individual is controlled, does not control the effect from the dataset size. For example, to rule out this, an experiment comparing two datasets, (1) 10k individuals and 1 biography for each individual (2) 2k individuals and 5 biographies for each individual will be required.
- In L358, the authors claim that the increase oscillation term can explain the advantage of replay methods in knowledge retention, but the connection between oscillation term and knowledge retention is unclear to me, and this is not directly confirmed by theoretical or experimental analysis.


W3: **On methodological design**- While the proposed method, STOC, can be effective in learning orthogonal knowledge during CPT,
- There is no further theoretical analysis that show this method can be strictly better compared to naive baselines or guarantees improved knowledge retention
- I think this design can be problematic when we want to overwrite previous knowledge in practical CPT scenarios, by the nature of the design. Specifically, if we want to continually update the model with new knowledge that conflicts with the previous one, STOC would generate replay data that contains previous knowledge that should be overwritten. I would like to hear from the authors on this potential issue.



[1] https://arxiv.org/abs/2305.16380

**Questions:**

Please see my questions in the weaknesses section.

---

> ### Author Response · Authors · 2025-11-27
> **Response to Review dSWz (Part 1)**
>
> Thanks for acknowledging our work and the constructive comments. Please kindly find point-to-point responses below.
> > **W1** Unclear technical novelty & contribution - Although the authors acknowledge they are building upon the 1-layer linear transformer constructions with strong assumptions used in previous works, what distincts this work from them remains unclear. I’d be happy to hear from authors about the novel points of each section and findings compared to previous insights. For example:
> ● Theorem 1 reflects the classical SGD dynamics on quadratic program, which is applied to the specific problem of CPT on linearized 1-layer Transformer.
> ● The theoretical and experimental result that attention patterns become more selective to subject tokens upon data augmentation can be inferred from the thorough analysis in [1], as they have shown the attention score for common tokens shrink at early phase of training, and data augmentation deliberately induces non-subject contexts become common.
>
> **Response:** Thanks for this insightful question.  Both the quadratic loss and the CrossEntropy loss exhibit similar gradient structures, which leads to results in Theorem 1 that resemble those for quadratic programs.  However, this does not imply that our conclusions are merely an application of quadratic programming. Our assumptions on data format, model architecture differ far from quadratic programming, and the resulting conclusions are better suitable to the process of FKA in LMs.
> We have carefully reviewed the reference you mentioned [1] and attempt to highlight several differences and contributions of our work. Although Theorem 3 in [1] also analyzes the attention score allocation behavior of the model, we note that the input texts we assume are completely different, and our problem is more complex. First, in our setting, a subject can correspond to multiple answers — for example, a person has multiple attributes, such as birthday and university. In [1], all tokens are defined as common tokens, with differences only in the degree of commonness. Meanwhile, Theorem 2 in [1] assumes only one common token and analyzes the effect on distinct tokens. Our analysis does not rely on such an assumption; of course, we make stronger assumptions about the model structure.
> In summary, we have restated our contribution as follows:
> 1. We have established a mathematical framework to analyze the training dynamics of LMs and therefore discover some unexplored phenomena.
> 2. We explore the cFKA process under non-zero initialization and the problem of learning new knowledge, covering two typical continual learning methods.
> 3. We not only validate our conclusions on multi-layer models but also propose a new Generative Data Replay method, which can be regarded as an extension of the theoretical results.
>
> [1] Scan and snap: Understanding training dynamics and token composition in 1-layer transformer, Neurips23.
>
> > **W2** Gap between the theory and experimental design
> ● In the current experimental protocol for data augmentation experiments, where the number of the biographies per individual is controlled, does not control the effect from the dataset size. For example, to rule out this, an experiment comparing two datasets, (1) 10k individuals and 1 biography for each individual (2) 2k individuals and 5 biographies for each individual will be required.
> ● In L358, the authors claim that the increase oscillation term can explain the advantage of replay methods in knowledge retention, but the connection between oscillation term and knowledge retention is unclear to me, and this is not directly confirmed by theoretical or experimental analysis.
>
> **Response:**  We appreciate your attention to the data augmentation experiment.  Number of biographies per individual  × individual num × epochs = training tokens. We control the number of tokens by adjusting epochs. Although the number of training epochs varies, we ensured all models reached convergence. Our setting follows previous work, as is commonly done in the literature[2-4].
> The second point you mentioned can be understood in the following way:
> 1. It's reasonable to assume $E[\xi(\tau)] = 0$ and $Cov[\xi(\tau)] = \Sigma$ when the time $t$ is large enough.
> 2. Let $A = Q\Lambda Q^{-1}$ and we have $Cov[z_s(t)] = Q[\sum_\tau \Lambda^{-\tau}\Sigma(Q^{-1})^\top\Lambda^{-\tau}]Q^{-1}$.
> 3. $P(s)\geq P(i) \propto f(Cov[z_s(t)])$ where f is monotonically increasing.
>
> Thus, one can think:  Larger amplitudes → greater historical impact → increased logits for low-frequency tokens → improved prediction accuracy (for low-frequency knowledge). In line 358, pretrained knowledge in the replay data has relatively low frequency given to the replay ratio.
>
> [2] Physics of Language Models: Part 3.1 Knowledge Storage and Extraction, ICML24.
> [3] Physics of Language Models: Part 3.3, Knowledge Capacity Scaling Laws, ICLR25.
> [4] Spurious Forgetting in Continual Learning of Language Models, ICLR25.

---

> ### Author Response · Authors · 2025-11-27
> **Response to Review dSWz (Part 2)**
>
> > **W3** On methodological design- While the proposed method, STOC, can be effective in learning orthogonal knowledge during CPT,
> ● There is no further theoretical analysis that show this method can be strictly better compared to naive baselines or guarantees improved knowledge retention
> ● I think this design can be problematic when we want to overwrite previous knowledge in practical CPT scenarios, by the nature of the design. Specifically, if we want to continually update the model with new knowledge that conflicts with the previous one, STOC would generate replay data that contains previous knowledge that should be overwritten. I would like to hear from the authors on this potential issue.
>
> **Response:** Thanks for the insightful questions.  We have to acknowledge that we are currently unable to provide a rigorous theoretical analysis for why STOC outperforms the baseline. Based on the theory and experimental results in Section 2, STOC can indeed identify knowledge fragments in sentences, such as subject information, according to attention scores. Naturally, using these knowledge fragments as prompts can increase the likelihood that the model generates pretrained knowledge. However, considering that the baseline methods involve additional data processing during pretraining, existing theoretical analyses cannot directly compare it with STOC.
> The second point describes a meaningful scenario, where conflicts exist between new and old knowledge. One possible technique relevant to this scenario is unlearning, which deliberately forces the model to forget certain old knowledge. We acknowledge that this paper does not analyze such scenarios. Instead, our main focus remains on catastrophic forgetting in continual training, where forgetting old knowledge is undesirable. In response to your question, we would like to restate the definition of CPT as follows.
> According to the technical reports/survey of popular open-source models [6-8], the training process of a (domain-specific) LLM typically consists of the following stages:
> 1. Pre-training: Training on a massive amount of tokens, e.g., Qwen3 is trained on 30T tokens.
> 2. Mid-training: Further training on domain-specific data to enrich the model’s scientific knowledge and reasoning ability. For instance, Qwen3 undergoes additional training on STEM and coding domains, amounting to roughly 5T tokens.
> 3. Domain-training: When the target is to build a domain-specific LLM (e.g., finance or biomedical), approximately 1T tokens of domain data are typically required.
> 4. Post-training: Usually includes instruction tuning and alignment stages.
> In this paper, we interpret both Stages 2 and 3 as part of the CPT phase, since their common objective is to inject new knowledge or capabilities into the model while preventing catastrophic forgetting.
>
> [6] Qwen3 Technical Report.
> [7] A Survey on LLM Mid-Training.
> [8] A Survey of Large Language Models in Finance (FinLLMs).

---

### Official Review · Reviewer_9Hi4 · 2025-10-31

**Soundness:** 2
**Presentation:** 1
**Contribution:** 2
**Rating:** 2
**Confidence:** 3

**Summary:**

The authors propose a theoretical framework that characterizes the training dynamics of cFKA using a single-layer Transformer with linear attention.
The analysis shows that regularization based methods only adjust the speed of convergence and not the final forgetting, but data replay methods can shift the convergence dynamics.
Based on the insight, STOC identifies influential factual snippets to improve upon regular replay methods.

**Strengths:**

- Analyzing forgetting through the lens of learning dynamics is an important direction; I'm glad to see papers making a push in this direction
- The separation of learning dynamics into Y and Z seem to be a clean choice for analysis
- The experiments on the synthetic Biograph shows significant benefit for STOC

**Weaknesses:**

- Style-wise, I find the paper is hard to follow:
	- The notation is a bit overwhelming and cluttered.
	- Some terms are introduced without proper explanation (e.g., oscillation amplitude) and can only get a better sense much later on, and wasn't clear to me whether large oscillation is desired or not.
- On a high-level, the studied setting seems to deviate from what is claimed. Specifically, the described setting in 2.1 is not really continued pretraining while in the paper the authors describe "[...] offering a unified explanation for the behavior of popular CPT methods". cFAK as described seems to be a very light post-training stage in terms of scale and characteristics.
- The effectiveness of STOC seems questionable. In Table 4, the perfomance of original is only 2-3 points above (often within 1 point) the baselines and given the number of datapoints in these datasets, it doesn't seem significant.
- Many smaller details seem off (mentioned in the question section).

**Questions:**

- The regularization coefficient in Table 2 is in the range of 1e6 to 1e8. If I'm not missing anything, this is unnaturally large. Can you explain why this is a justified choice?
- The dimension is off? Y \in R^{DxD} but D is supposed to be the vocab size?
- Eq 144: what's equation 6? There's no equation 6 in the main paper.
- What is Theorem 1 intended for? It's only showing a first-order tyler expansion of the error term, which is a dirivation step not a theorem. What is the assertion of the theorem?
- In Theorem 1, what is the matrix U and why is it relevant?

---

> ### Author Response · Authors · 2025-11-27
> **Response to Review 9Hi4 (Part 1)**
>
> Thanks for reviewing our work and the constructive comments. Please kindly find point-to-point responses below.
> > **W2** On a high level, the studied setting seems to deviate from what is claimed. Specifically, the described setting in 2.1 is not really continued pretraining while in the paper the authors describe "[...] offering a unified explanation for the behavior of popular CPT methods". cFKA, as described, seems to be a very light post-training stage in terms of scale and characteristics.
>
> **Response:** Thanks for your valuable feedback. We would like to restate the main logic of Section 2. We clarify that the framework established in Section 2.2 is designed to accommodate both PT and CPT, as the training procedures of PT and CPT are highly similar: they share the same training data format, model parameters, and loss function. The only difference is that PT starts from random initialization, whereas CPT is initialized with the parameter state at the end of PT. Before analyzing CPT, we first validate the applicability of the proposed mathematical framework in a PT-style simulation environment, demonstrating that our conclusions on simple models align with the experimental phenomena observed in multi-layer models. The analysis and verification under PT reflect the idea of cross-validation and lay the groundwork for the subsequent CPT analysis.
> We would also like to restate the positioning of CPT and cFKA, particularly regarding their training objectives and data scales. According to the technical reports/survey of popular open-source models [1-3], the training process of a (domain-specific) LLM typically consists of the following stages:
> 1. Pre-training: Training on a massive amount of tokens, e.g., Qwen3 is trained on 30T tokens.
> 2. Mid-training: Further training on domain-specific data to enrich the model’s scientific knowledge and reasoning ability. For instance, Qwen3 undergoes additional training on STEM and coding domains, amounting to roughly 5T tokens.
> 3. Domain-training: When the target is to build a domain-specific LLM (e.g., finance or biomedical), approximately 1T tokens of domain data are typically required.
> 4. Post-training: Usually includes instruction tuning and alignment stages.
>
> In this paper, we interpret both Stages 2 and 3 as part of the CPT phase, since their common objective is to inject new knowledge or capabilities into the model while preventing catastrophic forgetting. This phase typically involves billions of tokens and is highly resource-intensive. Because our work involves training multiple baselines, we initially use smaller real-world datasets in our experiments given the computational constraints.
> To address your concern, we have also made efforts to validate our proposed method on larger-scale real-world data.  Given the short discussion cycle and limited resources, we have done our best to add a larger-scale experiment for better empirical evidence.  Our settings and results are as follows.
> We first select `law_judiciary` subset of `IndustryCorpus2` as our CPT source, allowing us to scale domain-specific CPT data to 1B tokens. We additionally use `MMLU`, `MMLU-Redux-2.0`, and `SuperGPQA` as the evaluation benchmark, all containing QA pairs spanning more than 50 domains. The law subsets are used to assess how much new legal knowledge the model acquires during CPT, while the others are used to measure LMs' forgetting.
> As shown in the following table, STOC not only outperforms the baseline methods in mitigating catastrophic forgetting consistently, but also shows clear gains on the continual subsets. When scaling the continual pretraining data to larger corpora, the improvements remain stable rather than diminishing, indicating its robustness under larger-scale training.
> Method | MMLU Original | MMLU Continual | MMLU-Redux-2.0 Original | MMLU-Redux-2.0 Continual | SuperGPQA Original | SuperGPQA Continual |
> --------|---------------|----------------|-------------------------|--------------------------|------------------|-------------------|
>  Naive  | 22.49         | 24.40          | 21.93                   | 24.39                    | 9.48             | 10.98             |
> LAMOL  | 38.87         | 29.42          | 39.04                   | 28.05                    | 10.60            | 13.87             |
> STOC   | **40.17**     | **30.92**      | **40.26**               | **32.93**                | **10.76**        | **15.24**         |
>
> Inspired by your constructive comment, we have added explanatory text at the beginning of Section 2 in the main paper and included a schematic illustration (Figure 1). We also added more explanation of CPT in the introduction part to make the concept clearer. The larger-scale experiment results have been incorporated into the latest version of the paper, too. Thanks again for the constructive comments.
> [1] Qwen3 Technical Report.
> [2] A Survey on LLM Mid-Training.
> [3] A Survey of Large Language Models in Finance (FinLLMs).

---

> > ### Author Response · Authors · 2025-11-27
> > **Response to Review 9Hi4 (Part 2)**
> >
> > > **W1** Style-wise, I find the paper is hard to follow: The notation is a bit overwhelming and cluttered. Some terms are introduced without proper explanation (e.g., oscillation amplitude) and can only get a better sense much later on, and wasn't clear to me whether large oscillation is desired or not.
> >
> > **Response:** Thanks for your kind reminder. We think these symbols are all necessary to support the main conclusions in this paper, and we have tried our best to avoid unnecessary confusion. In response to your question, we have taken the following steps to improve your reading experience:
> > 1. We have added a Notation Table in the updated version (Appendix B), providing explanations for the main symbols, including their shapes and meanings.
> > 2. The second term on the right-hand side in Equation 1 is called the oscillation term since Eq 1. share the same format with damped oscillators. This term, naturally inherited from SGD training, can serve as a reminder to help LM remember training samples, especially those with low frequency. For example, if ``John'' rarely appears in the training corpus, then a bigger $\lambda_{\min}^+({H}_s)$ will lead to better memory for John-related knowledge. Thus, in the context of data replay with a large replay ratio, original knowledge can be better retained when the oscillation amplitude is relatively larger. Inspired by your question, we provide further clarification on the oscillation amplitude, which has been included in the updated version (page 4).
> > We appreciate the insights from your comments to help us polish our work.
> >
> > > **W3** The effectiveness of STOC seems questionable. In Table 4, the performance of the original is only 2-3 points above (often within 1 point) the baselines, and given the number of datapoints in these datasets, it doesn't seem significant.
> >
> > **Response:** Thanks for your insightful comments. We acknowledge that the improvement is not so large. In addition to the issue of training dataset size, we believe that the evaluation protocol is also a major factor. Since the KnowEdit benchmark does not specify which tokens correspond to 'knowledge' rather than 'template', the evaluation must rely on computing CrossEntropy over the full answer. This includes many low-information tokens, such as punctuation and prepositions, which dilute the measured gains.
> > In response to your comments, we have conducted multiple repeated experiments to compute the error bars for each method. The current results are based on 5 independent repetitions and can be further validated through permutation tests. We have provided the full table below, and these results have been incorporated into the latest version of the paper.
> > |       | Method            | ZSRE Original | ZSRE Continual | Wiki_Bio Original | Wiki_Bio Continual | Wiki_Recent Original | Wiki_Recent Continual |
> > |-------|-----------------|---------------|----------------|------------------|-------------------|--------------------|---------------------|
> > | Pythia | Naive            | 24.42 ± 0.27  | 48.48 ± 0.21   | 13.22 ± 0.12     | 32.21 ± 0.20      | 18.10 ± 0.21       | 20.39 ± 0.25        |
> > |       | LAMOL (α=0.5)     | 24.48 ± 0.22  | 47.56 ± 0.29   | 22.31 ± 0.15     | 31.33 ± 0.17      | 16.32 ± 0.21       | 19.27 ± 0.20        |
> > |       | LAMOL (α=0.8)     | 24.95 ± 0.32  | 47.12 ± 0.21   | 20.46 ± 0.20     | 31.54 ± 0.26      | 16.16 ± 0.24       | 17.35 ± 0.16        |
> > |       | STOC (α=0.5)      | 26.88 ± 0.23  | 47.94 ± 0.31   | 22.89 ± 0.24     | 28.05 ± 0.19      | 17.58 ± 0.13       | 19.23 ± 0.17        |
> > |       | STOC (α=0.8)      | **27.56 ± 0.20** | 47.23 ± 0.19 | **23.86 ± 0.10** | 31.88 ± 0.16      | **19.36 ± 0.13**   | 19.56 ± 0.19        |
> > | Qwen2.5 | Naive           | 34.58 ± 0.16  | 63.28 ± 0.28   | 32.33 ± 0.16     | 35.50 ± 0.13      | 19.28 ± 0.14       | 28.42 ± 0.18        |
> > |       | LAMOL (α=0.5)     | 37.54 ± 0.19  | 58.37 ± 0.22   | 31.29 ± 0.22     | 34.49 ± 0.23      | 20.48 ± 0.17       | 27.19 ± 0.21        |
> > |       | LAMOL (α=0.8)     | 36.71 ± 0.23  | 57.44 ± 0.17   | 34.67 ± 0.26     | 34.64 ± 0.17      | 20.15 ± 0.22       | 27.85 ± 0.19        |
> > |       | STOC (α=0.5)      | 37.12 ± 0.17  | 62.26 ± 0.12   | **35.57 ± 0.06** | 35.46 ± 0.10      | **21.40 ± 0.11**   | 28.75 ± 0.16        |
> > |       | STOC (α=0.8)      | **37.47 ± 0.14** | 62.59 ± 0.18 | 35.28 ± 0.13     | 33.16 ± 0.05      | 20.12 ± 0.15       | 27.34 ± 0.18        |
> >
> > Also,  the larger-scale evaluation on MMLU/MMLU-Redux/SuperGPQA (mention in the previous) will also alleviate your concerns. Again we are grateful for your constructive comments.

---

> > > ### Author Response · Authors · 2025-11-27
> > > **Response to Review 9Hi4 (Part 3)**
> > >
> > > > **Q1** The regularization coefficient in Table 2 is in the range of 1e6 to 1e8. If I'm not missing anything, this is unnaturally large. Can you explain why this is a justified choice?
> > >
> > > **Response:** We appreciate your attention to detail for the main process of regularization. Clearly, the larger the regularization coefficient, the less catastrophic forgetting one would expect. Our results precisely show that even such an unnaturally large regularization coefficient still fails to effectively mitigate catastrophic forgetting.  Our experimental experience shows that the method only becomes different (with naive pretraining) once the regularization reaches this scale. We additionally remark that 1e6 to 1e8 is the commonly accepted setting in the community[4].
> > > [4] Overcoming catastrophic forgetting in neural networks, Neurips17.
> > > [5] Spurious Forgetting in Continual Learning of Language Models, ICLR25.
> > >
> > > > **Q2** The dimension is off? Y \in R^{DxD} but D is supposed to be the vocab size?
> > >
> > > **Response:**  Yes, it refers to the vocab size, which is exactly how reparameterization trick works. it is a commonly-used approach and we list some papers to support[6,7].  In response to your concern, we have added a Notation Table in Appendix B of the paper, providing explanations for the main symbols, including their shapes and meanings.
> > > [6] Scan and snap: Understanding training dynamics and token composition in 1-layer transformer, Neurips23.
> > > [7] A Mathematical Framework for Transformer Circuits, Anthropic.
> > >
> > > > **Q3** Eq 144: what's equation 6? There's no equation 6 in the main paper.
> > >
> > > **Response:** Equation 6 is simply Equation 1 renumbered in the appendix. The hyperlink labels may have been duplicated. We apologize for the confusion.
> > >
> > > > **Q4** What is Theorem 1 intended for? It's only showing a first-order Taylor expansion of the error term, which is a derivation step, not a theorem. What is the assertion of the theorem?
> > >
> > > **Response:**  Theorem 1 characterizes both the convergence point and the convergence rate of the parameter matrix $Y$. First, it shows that Y converges toward a neighborhood of $U$, and that the convergence rate is exponential. Moreover, Theorem 1 states that $Y$ will eventually exhibit periodic motion within this neighborhood around $U$, a phenomenon that naturally arises from SGD. Although the proof only involves a Taylor expansion and solving differential equation, it provides crucial guidance for the subsequent analysis and represents one of the core conclusions of this paper.
> > >
> > > > **Q5** In Theorem 1, what is the matrix U and why is it relevant?
> > >
> > > **Response:** In Theorem1, $Y$ converges to a neighborhood around $U$. Once $Y$ has converged, we can approximate $Y$ with $U$ for further analysis.

---

### Official Review · Reviewer_obci · 2025-11-01

**Soundness:** 2
**Presentation:** 3
**Contribution:** 2
**Rating:** 4
**Confidence:** 3

**Summary:**

They study why continual pretraining makes language models forget old facts. They build a theoretical framework using a one-layer transformer to analyze how models acquire and retain factual knowledge. They show regularization only slows forgetting, while data replay stabilizes previous knowledge. Based on this, the authors propose STOC, a generative replay method selecting important tokens by attention. Their experiments confirm the theory and show STOC reduces catastrophic forgetting.

**Strengths:**

- They have clear motivation and propose a unified theoretical framework that formalizes continual factual knowledge acquisition (cFKA) as an analyzable training dynamic process.
- They clearly distinguishes how regularization and replay differ in convergence rate versus convergence point.
- They also validated on both synthetic and real datasets

**Weaknesses:**

- The theoretical analysis relies on a single-layer linear-attention Transformer, which may oversimplify the mechanisms observed in real multi-layer nonlinear architectures. While this abstraction is useful for interpretability, it remains unclear whether the same convergence dynamics generalize to the larger LMs.
- I think the real-data experiments are relatively limited in scope. Including larger-scale continual pre-training results or more diverse domains would strengthen the empirical support for the proposed claims.
-  Also, the current framework focuses mainly on templated and independent factual triplets. It would be valuable to discuss how the theory might extend to more complex knowledge structures, which are common in real-world factual updates.

**Questions:**

Do you think adapt replay ratio dynamically to task difficulty further mitigate forgetting?
Are results in Tables 3-4 statistically tested? How many random seeds? Are STOC's improvements within error margins?

---

> ### Author Response · Authors · 2025-11-27
> **Response to Review obci (Part 1)**
>
> Thanks for acknowledging our work and the constructive comments. Please kindly find point-to-point responses below.
> > **W1** The theoretical analysis relies on a single-layer linear-attention Transformer, which may oversimplify the mechanisms observed in real multi-layer nonlinear architectures. While this abstraction is useful for interpretability, it remains unclear whether the same convergence dynamics generalize to the larger LMs.
>
> **Response:**  Thanks for this insightful comment. We have to acknowledge that a rigorous theoretical analysis of parameter convergence in multi-layer models is beyond the scope of the current work. Such simplification is what we have to make to obtain meaningful results.
> (1) For multi-layer LMs, it is easy to see that the interactions across layers introduce significant complexity, making it extremely challenging or even untractable to provide a rigorous analysis. In fact, the vast majority of the literature focuses on single-layer models as the primary objects of study[1-5]. Some potential expansions of the theoretical analysis, e.g. positional embeddings and multi-head attention, are discussed in Appendix C to bridge the gap between theoretical and practical LMs.
> (2) Despite the theoretical difficulties, we have attempted to validate the effectiveness of our analysis through empirical experiments. All the experiments reported in this paper are conducted on multi-layer models, including the 160M Pythia (12 layers) and 0.5B Qwen2.5 (24 layers). The phenomena we observe are largely consistent with our theoretical predictions, suggesting that the insights from our analysis may generalize beyond the simplified single-layer setting.
> [1] Scan and snap: Understanding training dynamics and token composition in 1-layer transformer, Neurips23.
> [2] Transformers learn to implement preconditioned gradient descent for in-context learning, Neurips23.
> [3] How transformers learn causal structure with gradient descent, ICML24.
> [4] Trained transformers learn linear models in-context, JMLR24.
> [5] Understanding factual recall in transformers via associative memories, ICLR25.
>
> > **W2** I think the real-data experiments are relatively limited in scope. Including larger-scale continual pre-training results or more diverse domains would strengthen the empirical support for the proposed claims.
>
> **Response:** Thanks for your kind suggestions that can help us better revise our paper to be more solid.  Given the short discussion cycle and limited resources, we have done our best to add a larger-scale experiment for better empirical evidence.  Our settings and results are as follows.
> We first select `law_judiciary` subset of `IndustryCorpus2` as our CPT source, allowing us to scale domain-specific CPT data to 1B tokens. We additionally use `MMLU`, `MMLU-Redux-2.0`, and `SuperGPQA` as the evaluation benchmark, all containing QA pairs spanning more than 50 domains. The law subsets are used to assess how much new legal knowledge the model acquires during CPT, while the others are used to measure LMs' forgetting.
> As shown in the following table, STOC not only outperforms the baseline methods in mitigating catastrophic forgetting consistently, but also shows clear gains on the continual subsets. A plausible explanation is that replay stabilizes the model's internal representations by maintaining exposure to previously learned distributions. When scaling the continual pretraining data to larger corpora, the improvements remain stable rather than diminishing, indicating its robustness under larger-scale training.
>  Method | MMLU Original | MMLU Continual | MMLU-Redux-2.0 Original | MMLU-Redux-2.0 Continual | SuperGPQA Original | SuperGPQA Continual |
> --------|---------------|----------------|-------------------------|--------------------------|------------------|-------------------|
>  Naive  | 22.49         | 24.40          | 21.93                   | 24.39                    | 9.48             | 10.98             |
> LAMOL  | 38.87         | 29.42          | 39.04                   | 28.05                    | 10.60            | 13.87             |
> STOC   | **40.17**     | **30.92**      | **40.26**               | **32.93**                | **10.76**        | **15.24**         |
>
> We have included these results in the updated version (Page 10 in the main paper). Thanks again for your constructive feedback.

---

> > ### Author Response · Authors · 2025-11-27
> > **Response to Review obci (Part 2)**
> >
> > > **W3** The current framework focuses mainly on templated and independent factual triplets. It would be valuable to discuss how the theory might extend to more complex knowledge structures, which are common in real-world factual updates.
> >
> > **Response:** Thank you for raising this important question. This is one of the assumptions of this work.  The purpose of introducing templates is to ensure that each training sample contains only one piece of knowledge, which is purely for theoretical clarity and ease of analysis. Such simplification are commonly used in both theoretical[6] and empirical[7-9] works.
> > We would like to discuss the potential expansion of the knowledge modeling. As discussed in Finding 1, the factual knowledge is decomposed into frequency-based information and stored at the token level. In this case, the structure of fixed templates is very similar to more diverse knowledge expressions.  For example, as long as the knowledge can be expressed as an $k$-length tuple, the model can memorize it.
> > We also provide some empirical supports for the knowledge formulation. (1) Existing works[8] have shown that even without fully fixed templates, LMs can still acquire all knowledge from the training data. (2) In this paper, we use KnowEdit benchmark to simulate real-world factual updates, and MMLU/MMLU-Redux/SuperGPQA (added in discussion time) benchmark to simulate domain-specific knowledge. Answering these questions requires complicated, highly non-structured knowledge rather than simple factual triples. The results also demonstrate that LMs are still capable of learning new knowledge.
> > We appreciate your question, which highlights a critical aspect of potential expansion for this paper. We have expanded on this discussion in the updated version (Appendix D).
> > [6] Understanding factual recall in transformers via associative memories, ICLR25.
> > [7] A Survey on Knowledge Graphs: Representation, Acquisition, and Applications, IEEE19.
> > [8] Physics of Language Models: Part 3.1 Knowledge Storage and Extraction, ICML24.
> > [9] Spurious forgetting in continual learning of language models, ICLR25.
> >
> > > **Q1** Do you think adapting the replay ratio dynamically to task difficulty further mitigates forgetting?  Are the results in Tables 3-4 statistically tested? How many random seeds? Are STOC's improvements within error margins?
> >
> > **Response:** Thanks for your insightful comments.
> > Dynamically adjusting the replay ratio is a meaningful task and could be a potential extension of this work. However, this endeavor requires in-depth analysis and extensive engineering efforts, so we can only regard it as a direction for future research. Intuitively, the independence between new and old knowledge can provide guidance for selecting the replay ratio.
> > In response to your comments, we have conducted multiple repeated experiments to compute the error bars for each method. The current results are based on 5 independent repetitions and can be further validated through permutation tests. We have provided the full table below, and these results have been incorporated into the latest version of the paper.
> >
> > |       | Method            | ZSRE Original | ZSRE Continual | Wiki_Bio Original | Wiki_Bio Continual | Wiki_Recent Original | Wiki_Recent Continual |
> > |-------|-----------------|---------------|----------------|------------------|-------------------|--------------------|---------------------|
> > | Pythia | Naive            | 24.42 ± 0.27  | 48.48 ± 0.21   | 13.22 ± 0.12     | 32.21 ± 0.20      | 18.10 ± 0.21       | 20.39 ± 0.25        |
> > |       | LAMOL (α=0.5)     | 24.48 ± 0.22  | 47.56 ± 0.29   | 22.31 ± 0.15     | 31.33 ± 0.17      | 16.32 ± 0.21       | 19.27 ± 0.20        |
> > |       | LAMOL (α=0.8)     | 24.95 ± 0.32  | 47.12 ± 0.21   | 20.46 ± 0.20     | 31.54 ± 0.26      | 16.16 ± 0.24       | 17.35 ± 0.16        |
> > |       | STOC (α=0.5)      | 26.88 ± 0.23  | 47.94 ± 0.31   | 22.89 ± 0.24     | 28.05 ± 0.19      | 17.58 ± 0.13       | 19.23 ± 0.17        |
> > |       | STOC (α=0.8)      | **27.56 ± 0.20** | 47.23 ± 0.19 | **23.86 ± 0.10** | 31.88 ± 0.16      | **19.36 ± 0.13**   | 19.56 ± 0.19        |
> > | Qwen2.5 | Naive           | 34.58 ± 0.16  | 63.28 ± 0.28   | 32.33 ± 0.16     | 35.50 ± 0.13      | 19.28 ± 0.14       | 28.42 ± 0.18        |
> > |       | LAMOL (α=0.5)     | 37.54 ± 0.19  | 58.37 ± 0.22   | 31.29 ± 0.22     | 34.49 ± 0.23      | 20.48 ± 0.17       | 27.19 ± 0.21        |
> > |       | LAMOL (α=0.8)     | 36.71 ± 0.23  | 57.44 ± 0.17   | 34.67 ± 0.26     | 34.64 ± 0.17      | 20.15 ± 0.22       | 27.85 ± 0.19        |
> > |       | STOC (α=0.5)      | 37.12 ± 0.17  | 62.26 ± 0.12   | **35.57 ± 0.06** | 35.46 ± 0.10      | **21.40 ± 0.11**   | 28.75 ± 0.16        |
> > |       | STOC (α=0.8)      | **37.47 ± 0.14** | 62.59 ± 0.18 | 35.28 ± 0.13     | 33.16 ± 0.05      | 20.12 ± 0.15       | 27.34 ± 0.18        |

---

### Author Response · Authors · 2025-12-03
**Rebuttal Summary**

We sincerely thank all reviewers and chairs for devoting their time and effort to raise constructive comments and suggestions. To support assessment, we provide the following summary:
This paper studies the training dynamics of continual pretraining, which may include mid-training and domain adaptation phase. We present a theoretical framework that characterizes the training dynamics of a single-layer Transformer, offering a unified explanation for the behavior of popular CPT methods. Our analysis reveals that regularization-based methods fail in altering the inherent forgetting tendency, whereas data replay methods shift convergence dynamics and stabilize pretrained knowledge. Building on these insights, we propose a novel generative data replay approach, which identifies influential factual snippets to guide replay generation. We conduct extensive experiments to validate the theoretical findings and proposed method.

The reviewers acknowledged the importance of analyzing the CPT process and agreed that our theoretical analysis provides explanatory power for several empirical observations. However, they also suggested several possible directions for improvement:
1. The experiments on real-world datasets were considered insufficient in scale. The reviewers suggested adding significance tests or further validating the effectiveness of our proposed method on larger and more diverse datasets.
2. Some terms are introduced without proper explanation (e.g., oscillation amplitude). There should be additional clarification to make the meaning of each notation more explicit.
3. Our theoretical model might be too simplified to interpret modern language models. Specifically, some reviews thought discussions on diverse input structures, MLP layers, or multi-layer models were desired to make the results more practical.

Inspired by the feedback, we have made the following major revisions during rebuttal:
1. We repeated the original real-data experiments to obtain error bars, and additionally conducted experiments on a larger real-world dataset with general benchmarks. These supplementary results strengthen the credibility of our proposed method and further clarify the definition and goals of the CPT setting.
2. We provided more detailed explanations for several key concepts in the paper, such as the convergence point and amplitude. We added clarifications regarding the purpose and implications of the PT experiments in Section 2. We also added a notation table in the appendix for ease of reference. These improvements make the manuscript easier to follow.
3. Motivated by the comments, we expanded our discussion of potential extensions of the theoretical analysis, including adding MLP layers, analyzing multi-layer models,  different knowledge representation formats, and so on.  These extensions enrich the content of this paper and further demonstrate the flexibility of our proposed framework.

For further details, please refer to the discussion section and the updated submission. We once again thank all reviewers for their time and thoughtful feedback.

---

### Meta-Review · Area_Chair_eb6r · 2025-12-21

**Summary:**

The paper presents a theoretical framework that characterizes the training dynamics of a single-layer Transformer, offering a unified explanation for the behavior of popular CPT methods: regularization and data replay. The analysis indicates that regularization-based techniques do not change the natural tendency to forget, while data replay approaches modify the convergence behavior and help preserve pretrained knowledge.

**Reviewer Concerns:**

• Considering LLM as a single layer of Transformers may be too simple.

• Based on the scale of the dataset, the studied setting appears to deviate from what is claimed. The CPT phase described in the paper appears to be a light post-training stage in terms of scale and characteristics.

• The notions of the paper are hard to read.

• The improvement of STOC is limited and lacks theoretical guarantees.

**Reviewer Scores:**

Some concerns have been addressed by the authors during the rebuttal period, such as adding notation table and error bars. Moreover, most STOC-related experiments are conducted with freezing layers, yet the authors do not explain the rationale behind this choice.

The paper received ratings of 4, 2, 2, and 4. Although some reviewers expressed concerns about the simplicity of the theoretical model, I believe the abstraction is reasonable and largely aligned with prior work. As such, the model is sufficiently capable of capturing and explaining the key training dynamics during CPT.

However, the practical effectiveness of the proposed STOC method is limited, and its behavior may raise concerns in realistic CPT scenarios that involve conflicting or evolving knowledge. The additional experiments provided during the rebuttal do not fully address the core issues surrounding STOC’s design and applicability, as noted by reviewers obci, 9Hi4, and dSWz. Therefore, I don't think the reviewers would have increased their scores even if they had been able to fully participate in the discussion.

---

### Decision · Program_Chairs · 2026-01-26

Reject